# Removing Structured Noise using Diffusion Models

**Tristan S.W. Stevens**[1]                                    *t.s.w.stevens@tue.nl*

**Hans van Gorp**[1]                                           *h.v.gorp@tue.nl*

**Faik C. Meral**[2]                                           *can.meral@philips.com*

**Jun Seob Shin**[2]                                           *junseob.shin@philips.com*

**Jason Yu**[2]                                                *jason.yu@philips.com*

**Jean-Luc Robert**[1,2]                                       *jean-luc.robert@philips.com*

**Ruud J.G. van Sloun**[1]                                     *r.j.g.v.sloun@tue.nl*

[1] *Department of Electrical Engineering, Eindhoven University of Technology, The Netherlands*
[2] *Philips Research North America, Cambridge MA, USA*

**Reviewed on OpenReview:** *https://openreview.net/forum?id=BvKYsaOVEn*

## Abstract

Solving ill-posed inverse problems requires careful formulation of prior beliefs over the signals of interest and an accurate description of their manifestation into noisy measurements. Handcrafted signal priors based on e.g. sparsity are increasingly replaced by data-driven deep generative models, and several groups have recently shown that state-of-the-art score-based diffusion models yield particularly strong performance and flexibility. In this paper, we show that the powerful paradigm of posterior sampling with diffusion models can be extended to include rich, structured, noise models. To that end, we propose a joint conditional reverse diffusion process with learned scores for the noise and signal-generating distribution. We demonstrate strong performance gains across various inverse problems with structured noise, outperforming competitive baselines using normalizing flows, adversarial networks and various posterior sampling methods for diffusion models. This opens up new opportunities and relevant practical applications of diffusion modeling for inverse problems in the context of non-Gaussian measurement models.[1]

## 1 Introduction

Many signal and image processing problems, such as denoising, compressed sensing, or phase retrieval, can be formulated as inverse problems that aim to recover unknown signals from (noisy) observations. These ill-posed problems are, by definition, subject to many solutions under the given measurement model. Therefore, prior knowledge is required for a meaningful and physically plausible recovery of the original signal. Bayesian inference through posterior sampling incorporates both signal priors and observation likelihood models. Choosing an appropriate statistical prior is not trivial and dependent on both the application as well as the recovery task.

---

[1]**Code:** `https://github.com/tristan-deep/joint-diffusion`

In these image recovery tasks, the choice of noise prior is often assumed to be Gaussian or Poisson due to their mathematical tractability and ease of modeling. Corruptions in many applications, however, are often highly structured and spatially correlated. Therefore, besides accurate knowledge of the signal distribution, it is crucial to model the noise effectively. While it is often challenging to derive analytical models for these structured noise distributions, samples can be practically obtained through simulation or by isolating noise in the absence of the signal of interest. Relevant examples of structured noise include speckle, haze or interference. In medical imaging, for instance, ultrasound images are often corrupted by speckle noise, which limits contrast and complicates diagnoses (Yang et al., 2016). In computer vision, haze, fog and rain are highly correlated across neighboring pixels and can significantly degrade the quality of images. (Berman et al., 2016; Ren et al., 2019). Another example is the presence of interference in radar, which can lead to severe artifacts in the reconstructed range-Doppler maps (Uysal, 2018).

A popular approach for solving such problems involves Bayesian inference and inverse modeling, which requires the design of suitable priors. Before the advent of deep learning, sparsity in some transformed domain has been the go-to prior, such as iterative thresholding (Beck & Teboulle, 2009) or wavelet decomposition (Mallat, 1999). At present, deep generative modeling has established itself as a strong mechanism for learning such priors for inverse problem-solving. Both generative adversarial networks (GANs) (Bora et al., 2017) and normalizing flows (NFs) (Asim et al., 2020; Wei et al., 2022) have been applied as natural signal priors for inverse problems in image recovery. These data-driven methods are more powerful compared to classical methods, as they can accurately learn the natural signal manifold and do not rely on assumptions such as signal sparsity or hand-crafted basis functions.

Recently, diffusion models have shown impressive results for both conditional and unconditional image generation and can be easily fitted to a target data distribution using score matching (Song et al., 2020). These deep generative models learn the score of the data manifold and produce samples by reverting a diffusion process, guiding noise samples toward the target distribution. Diffusion models have achieved state-of-the-art performance in many downstream tasks and applications, ranging from state-of-the-art text-to-image models such as Stable Diffusion (Rombach et al., 2022) to medical imaging (Song et al., 2021b; Jalal et al., 2021a; Chung & Ye, 2022). Furthermore, understanding of diffusion models is rapidly improving and progress in the field is extremely fast-paced (Chung et al., 2022b; Bansal et al., 2022; Daras et al., 2022a; Karras et al., 2022; Luo, 2022). The iterative nature of the sampling procedure used by diffusion models renders inference slow compared to GANs and VAEs. However, many recent efforts have shown ways to significantly improve the sampling speed by accelerating the diffusion process, from improving the sampling process itself (Salimans & Ho, 2021; Daras et al., 2022b; Chung et al., 2022c; Stevens et al., 2025; Park et al., 2024), to executing the diffusion in some reduced (latent) space (Jing et al., 2022; Vahdat et al., 2021; Rombach et al., 2022).

Despite this promise, current score-based diffusion methods for inverse problems are limited to measurement models with unstructured noise. In many image processing tasks, corruptions are however highly structured and spatially correlated. Nevertheless, current conditional diffusion models naively assume that the noise follows some basic tractable distribution (e.g. Gaussian or Poisson). Diffusion Posterior Sampling (DPS) (Chung et al., 2022a), Diffusion Model Based Posterior Sampling (DMPS) (Meng & Kabashima, 2022), and Pseudoinverse-guided Diffusion Models (ΠGDM) (Song et al., 2023), all have a different take on posterior sampling with diffusion models. Namely, they seek to approximate the intractable noise-perturbed likelihood score, usually involving Tweedie's formula, in various ways. RED-diff sidesteps the challenge of posterior score approximation using variational inference (Mardani et al., 2023), resulting in a simple gradient update rule that resembles regularization-by-denoising. Denoising Diffusion Restoration Models (DDRM) take another approach altogether by performing the diffusion trajectory in the spectral space, tying the measurement noise to the diffusion noise (Kawar et al., 2022). Albeit still under the Gaussian assumption. Denoising Diffusion Null-Space Models (DDNM) (Wang et al., 2022) opt for a different decomposition by projecting samples to the null-space of the forward operator of noiseless and noisy (Gaussian) inverse problems. Finally, Deep Equilibrium Diffusion Restoration (DeqIR) rethinks the sampling process by modeling it as a fixed point system, achieving faster parallel sampling (Cao et al., 2024). To summarize, all these methods improve upon incorporating measurements into the diffusion process. Nonetheless, they limit their scope to classic inverse problems such as denoising (Gaussian), inpainting, super-resolution, deblurring, etc., and do not address problems with structured noise. Luo et al. (2023) propose a general-purpose image restoration framework

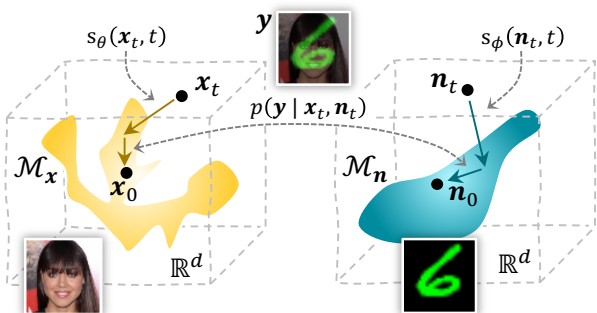

Figure 1: Overview of the proposed joint posterior sampling method for removing structured noise using diffusion models. During the sampling process, the solutions for both signal and noise move toward their respective data manifold $\mathcal{M}$ through score models $s_\theta$ and $s_\phi$. At the same time, the data consistency term derived from the joint likelihood $p(\boldsymbol{y}|\boldsymbol{x}_t, \boldsymbol{n}_t)$ ensures solutions that are in line with the (structured) noisy measurement $\boldsymbol{y} = \boldsymbol{A}\boldsymbol{x} + \boldsymbol{n}$.

for arbitrary degradations. Unfortunately, this requires *clean-noisy* sample pairs for training, leading to models that are task-specific, need retraining, and are more vulnerable to out-of-distribution data.

Beyond the realm of diffusion models, Whang et al. (2021) extended normalizing flow (NF)-based inference to structured noise applications. However, compared to diffusion models, NFs require specialized network architectures, which are computationally and memory expensive.

Given the promising outlook of diffusion models, we propose to learn score models for both the noise and the desired signal and perform joint inference of both quantities, coupled via the observation model. The resulting sampling scheme enables solving a wide variety of inverse problems with structured noise.

The main contributions of this work are as follows:

- We propose a novel joint conditional approximate posterior sampling method to efficiently remove structured noise using diffusion models. Our formulation is compatible with many existing iterative sampling methods for score-based generative models.

- We show strong performance gains across various challenging inverse problems involving structured noise compared to competitive state-of-the-art methods based on NFs, GANs, and diffusion models.

- We provide derivations for and comparison of three recent posterior sampling frameworks for diffusion models (ΠGDM, DPS, projection) as the backbone for our joint inference scheme.

- We demonstrate improved robustness on a range of out-of-distribution signals and noise compared to baselines.

## 2 Problem Statement

Many image reconstruction tasks can be formulated as an inverse problem with the basic form $\boldsymbol{y} = \boldsymbol{A}\boldsymbol{x} + \boldsymbol{n}$, where $\boldsymbol{y} \in \mathbb{R}^m$ is the noisy observation, $\boldsymbol{x} \in \mathbb{R}^d$ the desired signal or image, and $\boldsymbol{n} \in \mathbb{R}^m$ the additive noise. The linear forward operator $\boldsymbol{A} \in \mathbb{R}^{m \times d}$ captures the deterministic transformation of $\boldsymbol{x}$. Maximum a posteriori (MAP) inference is typically used to find an optimal solution $\hat{\boldsymbol{x}}_{\text{MAP}}$ that maximizes posterior density $p_{X|Y}(\boldsymbol{x}|\boldsymbol{y})$:

$$\hat{\boldsymbol{x}}_{\text{MAP}} = \arg\max_{\boldsymbol{x}} \log p_{X|Y}(\boldsymbol{x}|\boldsymbol{y}) = \arg\max_{\boldsymbol{x}} \left[\log p_{Y|X}(\boldsymbol{y}|\boldsymbol{x}) + \log p_X(\boldsymbol{x})\right], \tag{1}$$

where $p_{Y|X}(\boldsymbol{y}|\boldsymbol{x})$ is the likelihood according to the measurement model and $\log p_X(\boldsymbol{x})$ the signal prior. Assumptions on the stochastic corruption process $\boldsymbol{n}$ are of key importance too, in particular for applications for which this process is highly structured. However, most methods assume i.i.d. Gaussian distributed noise, such that the forward model becomes $p_{Y|X}(\boldsymbol{y}|\boldsymbol{x}) \sim \mathcal{N}(\boldsymbol{A}\boldsymbol{x}, \sigma_N^2 \mathbf{I})$. This naturally leads to the following simplified problem:

$$\hat{\boldsymbol{x}}_{\text{MAP}} = \arg\min_{\boldsymbol{x}} \frac{1}{2\sigma_N^2} ||\boldsymbol{y} - \boldsymbol{A}\boldsymbol{x}||_2^2 - \log p_X(\boldsymbol{x}). \tag{2}$$

However, this naive assumption can be very restrictive as many noise processes are much more structured and complex. A myriad of problems can be addressed under the assumed measurement model, given the freedom of choice for the noise source $\boldsymbol{n}$. Therefore, in this work, our aim is to solve a more broad class of inverse problems defined by any arbitrary noise distribution $\boldsymbol{n} \sim p_N(\boldsymbol{n}) \neq \mathcal{N}$ and signal prior $\boldsymbol{x} \sim p_X(\boldsymbol{x})$, resulting in the following, more general, MAP estimator:

$$\hat{\boldsymbol{x}}_{\text{MAP}} = \arg\max_{\boldsymbol{x}} \log p_N(\boldsymbol{y} - \boldsymbol{A}\boldsymbol{x}) + \log p_X(\boldsymbol{x}). \tag{3}$$

In this paper, we propose to solve this class of problems using flexible diffusion models. Moreover, diffusion models naturally enable posterior sampling, i.e. $\boldsymbol{x} \sim p_{X|Y}(\boldsymbol{x}|\boldsymbol{y})$, allowing us to take advantage of the benefits thereof (Jalal et al., 2021b; Kawar et al., 2021; Daras et al., 2022a) with respect to the MAP estimator which simply collapses the posterior distribution into a single point estimate.

## 2.1 Background

Score-based diffusion models have been introduced independently as score-based models (Song & Ermon, 2019; 2020) and denoising diffusion probabilistic modeling (DDPM) (Ho et al., 2020). In this work, we will consider the formulation introduced by Song et al. (2020), which unifies both perspectives on diffusion models by expressing diffusion as a continuous-time process through stochastic differential equations (SDE). Diffusion models produce samples by reversing a corruption (noising) process. In essence, these models are trained to denoise their inputs for each timestep in the corruption process. Through iteration of this reverse process, samples can be drawn from a learned data distribution, starting from random noise.

The diffusion process of the data $\{\boldsymbol{x}_t \in \mathbb{R}^d\}_{t \in [0,1]}$ is characterized by a continuous sequence of Gaussian perturbations of increasing magnitude indexed by time $t \in [0,1]$. Starting from the data distribution at $t = 0$, clean images are defined by $\boldsymbol{x}_0 \sim p(\boldsymbol{x}_0) \equiv p(\boldsymbol{x})$. Forward diffusion can be described using an SDE as follows: $d\boldsymbol{x}_t = f(t)\boldsymbol{x}_t dt + g(t)d\mathbf{w}$, where $\boldsymbol{w} \in \mathbb{R}^d$ is a standard Wiener process, $f(t) : [0,1] \to \mathbb{R}$ and $g(t) : [0,1] \to \mathbb{R}$ are the drift and diffusion coefficients, respectively. Moreover, these coefficients are chosen so that the resulting distribution $p(\boldsymbol{x}_1)$ at the end of the perturbation process approximates a predefined base distribution $p(\boldsymbol{x}_1) \approx \pi(\boldsymbol{x}_1)$. Furthermore, the transition kernel of the diffusion process can be defined in one step as $q(\boldsymbol{x}_t|\boldsymbol{x}_0) \sim \mathcal{N}(\boldsymbol{x}_t|\alpha_t \boldsymbol{x}_0, \beta_t^2 \mathbf{I})$, where $\alpha_t$ and $\beta_t$ can be analytically derived from the SDE.

Naturally, we are interested in reversing the diffusion process, so that we can sample from $\boldsymbol{x}_0 \sim p(\boldsymbol{x}_0)$. The reverse diffusion process is also a diffusion process given by the reverse-time SDE (Anderson, 1982; Song et al., 2020):

$$d\boldsymbol{x}_t = \left\{ f(t)\boldsymbol{x}_t - g(t)^2 \underbrace{\nabla_{\boldsymbol{x}_t} \log p(\boldsymbol{x}_t)}_{\text{score}} \right\} dt + g(t)d\bar{\boldsymbol{w}}_t \tag{4}$$

where $\bar{\boldsymbol{w}}_t$ is the standard Wiener process in the reverse direction. The gradient of the log-likelihood of the data with respect to itself, a.k.a. the *score function*, arises from the reverse-time SDE. The score function is a gradient field pointing back to the data manifold and can intuitively be used to guide a random sample from the base distribution $\pi(\boldsymbol{x})$ to the desired data distribution. Given a dataset $\mathcal{X} = \left\{ \boldsymbol{x}_0^{(1)}, \boldsymbol{x}_0^{(2)}, \ldots, \boldsymbol{x}_0^{(|\mathcal{X}|)} \right\} \sim p(\boldsymbol{x}_0)$, scores can be estimated by training a neural network $s_\theta(\boldsymbol{x}_t, t)$ parameterized by weights $\theta$, with score matching techniques such as the denoising score matching (DSM) objective (Vincent, 2011):

$$\theta^* = \arg\min_{\theta} \mathbb{E}_{t \sim U[0,1]} \left\{ \mathbb{E}_{(\boldsymbol{x}_0, \boldsymbol{x}_t) \sim p(\boldsymbol{x}_0)q(\boldsymbol{x}_t|\boldsymbol{x}_0)} \left[ ||s_\theta(\boldsymbol{x}_t, t) - \nabla_{\boldsymbol{x}_t} \log q(\boldsymbol{x}_t|\boldsymbol{x}_0)||_2^2 \right] \right\}. \tag{5}$$

Given a sufficiently large dataset $\mathcal{X}$ and model capacity, DSM ensures that the score network converges to $s_\theta(\boldsymbol{x}_t, t) \simeq \nabla_{\boldsymbol{x}_t} \log p(\boldsymbol{x}_t)$. After training the time-dependent score model $s_\theta$, it can be used to calculate the reverse-time diffusion process and solve the trajectory using numerical samplers such as the Euler-Maruyama algorithm. Alternatively, more sophisticated samplers, such as ALD (Song & Ermon, 2019), probability flow

ODE (Song et al., 2020), and Predictor-Corrector sampler (Song et al., 2020), can be used to further improve sample quality.

These iterative sampling algorithms discretize the continuous time SDE into a sequence of time steps $\{0 = t_0, t_1, \ldots, t_T = 1\}$, where a noisy sample $\hat{\boldsymbol{x}}_{t_i}$ is denoised to produce a sample for the next time step $\hat{\boldsymbol{x}}_{t_{i-1}}$. The resulting samples $\{\hat{\boldsymbol{x}}_{t_i}\}_{i=0}^T$ constitute an approximation of the actual diffusion process $\{\boldsymbol{x}_t\}_{t\in[0,1]}$.

## 3 Method

In this section, we outline our approach to solving inverse problems under structured noise. Section 3.1 introduces our joint posterior sampling framework, leveraging joint diffusion processes for signal and noise. Section 3.2 discusses data consistency rules, detailing different methods to ensure alignment with observations. Additionally, we demonstrate compatibility of our method with common posterior sampling strategies.

### 3.1 Joint Posterior Sampling under Structured Noise

We are interested in posterior sampling under structured noise. We recast this as a joint optimization problem with respect to the signal $\boldsymbol{x}$ and noise $\boldsymbol{n}$ given by:

$$(\boldsymbol{x}, \boldsymbol{n}) \sim p_{X,N}(\boldsymbol{x}, \boldsymbol{n}|\boldsymbol{y}) \propto p_{Y|X,N}(\boldsymbol{y}|\boldsymbol{x}, \boldsymbol{n}) \cdot p_X(\boldsymbol{x}) \cdot p_N(\boldsymbol{n}), \tag{6}$$

where we assume the signal and noise components to be independent. Solving inverse problems using diffusion models requires conditioning of the diffusion process on the observation $\boldsymbol{y}$, such that we can sample from the posterior $p_{X|Y}(\boldsymbol{x}, \boldsymbol{n}|\boldsymbol{y})$. Therefore, we construct a *joint conditional* diffusion process $\{\boldsymbol{x}_t, \boldsymbol{n}_t|\boldsymbol{y}\}_{t\in[0,1]}$, in turn producing a *joint conditional* reverse-time SDE:

$$\mathrm{d}(\boldsymbol{x}_t, \boldsymbol{n}_t) = \left\{ f(t)(\boldsymbol{x}_t, \boldsymbol{n}_t) - g(t)^2 \nabla_{\boldsymbol{x}_t, \boldsymbol{n}_t} \log p(\boldsymbol{x}_t, \boldsymbol{n}_t|\boldsymbol{y}) \right\} \mathrm{d}t + g(t)\mathrm{d}\bar{\mathbf{w}}_t. \tag{7}$$

We would like to factorize the posterior using our learned *unconditional* score model and tractable measurement model, given the joint formulation. Consequently, we construct two separate diffusion processes, defined by separate score models but entangled through the measurement model $p_{Y|X,N}(\boldsymbol{y}|\boldsymbol{x}, \boldsymbol{n})$. In addition to the original score model $s_\theta(\boldsymbol{x}, t)$, we introduce a second score model $s_\phi(\boldsymbol{n}_t, t) \simeq \nabla_{\boldsymbol{n}_t} \log p_N(\boldsymbol{n}_t)$, parameterized by weights $\phi$, to model the expressive noise component $\boldsymbol{n}$. These two score networks can be trained independently on datasets for $\boldsymbol{x}$ and $\boldsymbol{n}$, respectively, using the objective in equation 5. This is a significant differentiator, as our method eliminates the need to collect samples of signals and noise together with corresponding ground truth. Self-supervised generative modeling on isolated signals and noise measurements is sufficient, thus relaxing the difficulty of curating signal and noise datasets. The gradients of the posterior with respect to $\boldsymbol{x}$ and $\boldsymbol{n}$, used in equation 7, are now given by:

$$\nabla_{\boldsymbol{x}_t, \boldsymbol{n}_t} \log p(\boldsymbol{x}_t, \boldsymbol{n}_t|\boldsymbol{y}) = \begin{bmatrix} \nabla_{\boldsymbol{x}_t} \log p(\boldsymbol{x}_t, \boldsymbol{n}_t|\boldsymbol{y}) \\ \nabla_{\boldsymbol{n}_t} \log p(\boldsymbol{x}_t, \boldsymbol{n}_t|\boldsymbol{y}) \end{bmatrix} \approx \begin{bmatrix} s_\theta^*(\boldsymbol{x}_t, t) + \lambda \nabla_{\boldsymbol{x}_t} \log p(\boldsymbol{y}|\boldsymbol{x}_t, \boldsymbol{n}_t) \\ s_\phi^*(\boldsymbol{n}_t, t) + \kappa \nabla_{\boldsymbol{n}_t} \log p(\boldsymbol{y}|\boldsymbol{x}_t, \boldsymbol{n}_t) \end{bmatrix}, \tag{8}$$

which simply factorizes the joint posterior into prior and likelihood terms using Bayes' rule from equation 6 for both diffusion processes. Following the literature on classifier-(free) diffusion guidance (Dhariwal & Nichol, 2021; Ho & Salimans, 2022) and diffusion for inverse problems (Song et al., 2020; Chung et al., 2022a; Song et al., 2023), two Bayesian weighting terms, $\lambda$ and $\kappa$, are introduced. These terms are tunable hyperparameters that weigh the importance of following the prior, $s_\theta^*(\boldsymbol{x}_t, t)$ and $s_\phi^*(\boldsymbol{n}_t, t)$, versus the measurement model, $\nabla_{\boldsymbol{n}_t} \log p(\boldsymbol{y}|\boldsymbol{x}_t, \boldsymbol{n}_t)$. A conceptual overview of the proposed method is shown in Fig. 1.

### 3.2 Data Consistency Rules

The resulting *true* noise-perturbed likelihood $p(\boldsymbol{y}|\boldsymbol{x}_t, \boldsymbol{n}_t)$ is generally intractable, unlike $p(\boldsymbol{y}|\boldsymbol{x}_0, \boldsymbol{n}_0)$. Different approximations have been proposed in recent works (Song et al., 2020; Chung et al., 2022a; Song et al., 2023; Meng & Kabashima, 2022; Feng et al., 2023; Finzi et al., 2023). Our method is agnostic to the type of data-consistency rule employed. To study its effect on the final output, we will implement three strong approaches proposed in literature, namely, Pseudoinverse-Guided Diffusion Models (ΠGDM) (Song et al.,

Table 1: Parameter choices for the Gaussian model of the noise-perturbed likelihood function in equation 9.

| | ΠGDM (Song et al., 2023) | DPS (Chung et al., 2022a) | Projection (Song et al., 2020) |
|---|---|---|---|
| $\boldsymbol{\gamma}_t$ | $\boldsymbol{y}$ | $\boldsymbol{y}$ | $\hat{\boldsymbol{y}}_t$ |
| $\boldsymbol{\mu}_t$ | $A\boldsymbol{x}_{0|t} + \boldsymbol{n}_{0|t}$ | $A\boldsymbol{x}_{0|t} + \boldsymbol{n}_{0|t}$ | $A\boldsymbol{x}_t + \boldsymbol{n}_t$ |
| $\boldsymbol{\Sigma}_t$ | $\left(r_t^2 \boldsymbol{A}\boldsymbol{A}^\mathsf{T} + q_t^2 I\right)$ | $\rho^2 I$ | $\rho^2 I$ |
| $\lambda$ | $\lambda' r_t^2 / g(t)^2$ | $\lambda' \rho^2 / \left(g(t)^2 |\boldsymbol{y} - \mu|_2^1\right)$ | $\lambda' \rho^2 / g(t)^2$ |
| $\kappa$ | $\kappa' q_t^2 / g(t)^2$ | $\kappa' \rho^2 / \left(g(t)^2 |\boldsymbol{y} - \mu|_2^1\right)$ | $\kappa' \rho^2 / g(t)^2$ |

2023), Diffusion Posterior Sampling (DPS) (Chung et al., 2022a), and projection (Song et al., 2020) and see how they can be leveraged for our joint posterior sampling framework. In all methods, to ensure traceability of $p(\boldsymbol{y}|\boldsymbol{x}_t, \boldsymbol{n}_t)$, it is modeled as a Gaussian, namely:

$$p(\boldsymbol{y}|\boldsymbol{x}_t, \boldsymbol{n}_t) \approx \mathcal{N}(\boldsymbol{\gamma}_t; \boldsymbol{\mu}_t, \boldsymbol{\Sigma}_t), \tag{9}$$

where the three different methods employ different approximations for the parameters of the Normal distribution. In all three methods, the covariance $\boldsymbol{\Sigma}_t$ is not a function of $\boldsymbol{x}_t$ or $\boldsymbol{n}_t$, and we can thus write the noise-perturbed likelihood score as:

$$\nabla_{\boldsymbol{x}_t, \boldsymbol{n}_t} \log p(\boldsymbol{y}|\boldsymbol{x}_t, \boldsymbol{n}_t) \approx [\nabla_{\boldsymbol{x}_t, \boldsymbol{n}_t} \boldsymbol{\mu}_t] \boldsymbol{\Sigma}_t^{-1}(\boldsymbol{\gamma}_t - \boldsymbol{\mu}_t). \tag{10}$$

We will now specifically derive the sampling procedure for our joint diffusion process using ΠGDM as basis. Additionally, we provide derivations for DPS and the projection method in Appendix A. Finally, Table 1 shows an overview of the choices made for each parameter in the different methods.

Similar to the original ΠGDM paper, we start with an approximation of $\boldsymbol{x}_t, \boldsymbol{n}_t$ toward $\boldsymbol{x}_0, \boldsymbol{n}_0$, which then allows the usage of the known relationship of $p(\boldsymbol{y}|\boldsymbol{x}_0, \boldsymbol{n}_0)$. Since $\boldsymbol{y}$, $\boldsymbol{x}_t$, and $\boldsymbol{n}_t$ are conditionally independent given $\boldsymbol{x}_0$ and $\boldsymbol{n}_0$, we can write:

$$p(\boldsymbol{y}|\boldsymbol{x}_t, \boldsymbol{n}_t) = \int_{\boldsymbol{x}_0} \int_{\boldsymbol{n}_0} p(\boldsymbol{x}_0|\boldsymbol{x}_t)p(\boldsymbol{n}_0|\boldsymbol{n}_t)p(\boldsymbol{y}|\boldsymbol{x}_0, \boldsymbol{n}_0)\mathrm{d}\boldsymbol{n}_0\mathrm{d}\boldsymbol{x}_0, \tag{11}$$

which is a marginalization over $\boldsymbol{x}_0$ and $\boldsymbol{n}_0$. Now, we have substituted the intractability of computing $p(\boldsymbol{y}|\boldsymbol{x}_t, \boldsymbol{n}_t)$, for the intractability of computing (scores of) $p(\boldsymbol{x}_0|\boldsymbol{x}_t)$ and $p(\boldsymbol{n}_0|\boldsymbol{n}_t)$. ΠGDM then estimates $p(\boldsymbol{x}_0|\boldsymbol{x}_t)$ using variational inference (VI), where it models the reverse diffusion steps as Gaussians, which we extend here to the noise as well:

$$\begin{cases} p(\boldsymbol{x}_0|\boldsymbol{x}_t) \approx \mathcal{N}(\boldsymbol{x}_{0|t}, r_t^2 I) \\ p(\boldsymbol{n}_0|\boldsymbol{n}_t) \approx \mathcal{N}(\boldsymbol{n}_{0|t}, q_t^2 I), \end{cases} \tag{12}$$

where $q_t^2$ and $r_t^2$ represent the uncertainty or error made in the VI. The means of the Gaussian approximations $(\boldsymbol{x}_{0|t}, \boldsymbol{n}_{0|t})$ are calculated using Tweedie's formula, which can be thought of as a one-step denoising process using our trained diffusion model to estimate the *true* $\boldsymbol{x}_0$ and $\boldsymbol{n}_0$:

$$\boldsymbol{x}_{0|t} = \mathbb{E}[\boldsymbol{x}_0|\boldsymbol{x}_t] = \frac{\boldsymbol{x}_t + \beta_t^2 \nabla_{\boldsymbol{x}_t} \log p(\boldsymbol{x}_t)}{\alpha_t} \approx \frac{\boldsymbol{x}_t + \beta_t^2 s_\theta^*(\boldsymbol{x}_t, t)}{\alpha_t}, \tag{13}$$

with an analogous equation for $\boldsymbol{n}_{0|t}$. Here, $\alpha_t$ and $\beta_t$ can be derived from the SDE formulation as mentioned in Section 2.1. Substitution of the VI estimate (equation 12 into equation 11), then results in an approximation of the noise-perturbed likelihood:

$$p(\boldsymbol{y}|\boldsymbol{x}_t, \boldsymbol{n}_t) \approx \mathcal{N}(\boldsymbol{\gamma}_t; \boldsymbol{\mu}_t, \boldsymbol{\Sigma}_t) \begin{cases} \boldsymbol{\gamma}_t = \boldsymbol{y} \\ \boldsymbol{\mu}_t = A\boldsymbol{x}_{0|t} + \boldsymbol{n}_{0|t} \\ \boldsymbol{\Sigma}_t = r_t^2 \boldsymbol{A}\boldsymbol{A}^\mathsf{T} + q_t^2 I. \end{cases} \tag{14}$$

Subsequently, we derive the following estimated noise-perturbed likelihood scores:

$$\begin{bmatrix} \nabla_{\boldsymbol{x}_t} \log p(\boldsymbol{y}|\boldsymbol{x}_t, \boldsymbol{n}_t) \\ \nabla_{\boldsymbol{n}_t} \log p(\boldsymbol{y}|\boldsymbol{x}_t, \boldsymbol{n}_t) \end{bmatrix} \approx \begin{bmatrix} (\nabla_{\boldsymbol{x}_t}\boldsymbol{x}_{0|t})\ \boldsymbol{A}^\mathsf{T}\boldsymbol{\Sigma}_t^{-1}(\boldsymbol{y} - \boldsymbol{A}\boldsymbol{x}_{0|t} - \boldsymbol{n}_{0|t}) \\ (\nabla_{\boldsymbol{n}_t}\boldsymbol{n}_{0|t})\ \ \ \boldsymbol{\Sigma}_t^{-1}(\boldsymbol{y} - \boldsymbol{A}\boldsymbol{x}_{0|t} - \boldsymbol{n}_{0|t}) \end{bmatrix}, \tag{15}$$

where $\nabla_{\boldsymbol{x}_t}\boldsymbol{x}_{0|t}$ and $\nabla_{\boldsymbol{n}_t}\boldsymbol{n}_{0|t}$ are the Jacobians of equation 13, which can be computed using automatic differentiation methods. In ΠGDM, the Bayesian weighting terms $\lambda$ and $\kappa$ are not fixed scalars, rather these are chosen to be equal to the estimated VI variances, $r_t^2$ and $q_t^2$. Additionally, the diffusion coefficient $g(t)^2$ gets cancelled out in the weighting scheme. Lastly, in this work, we introduce the additional explicit scalars $\lambda'$ and $\kappa'$, to bring it in line with the other data consistency rules. Note that introducing these scalars is the same as scaling $r_t^2$ and $q_t^2$ by a fixed amount for all timesteps.

Song et al. (2023) provide recommendations for choosing the variance of the VI , namely $r_t^2 = \frac{\beta^2}{\beta^2-1}$, when the noise model is a known tractable distribution, which we adopt. Additionally, since we here introduce the notion of modeling $\boldsymbol{n}$ using a different diffusion model, we also set the variance of the VI estimate of $p(\boldsymbol{n}_0|\boldsymbol{n}_t)$ to $q_t^2 = r_t^2$, as it is subjected to a similar SDE trajectory.

---

**Algorithm 1:** Joint posterior sampling with ΠGDM for score-based diffusion models

**Require:** $T, s_\theta, s_\phi, \lambda, \kappa, r_t^2, q_t^2, \boldsymbol{y}$

1   $\boldsymbol{x}_T \sim \pi(\boldsymbol{x}), \boldsymbol{n}_1 \sim \pi(\boldsymbol{n}), \Delta t \leftarrow \frac{1}{T}$

2

3   **for** $i = T-1$ **to** $0$ **do**

4     $t \leftarrow \frac{i+1}{T}$

     // Data consistency steps

5     $\boldsymbol{x}_{0|t} \leftarrow (\boldsymbol{x}_t + \beta_t^2 s_\theta^*((\boldsymbol{x}_t,t))/\alpha_t$

6     $\boldsymbol{n}_{0|t} \leftarrow (\boldsymbol{n}_t + \beta_t^2 s_\phi^*((\boldsymbol{n}_t,t))/\alpha_t$

7     $\boldsymbol{\mu}_t \leftarrow \boldsymbol{A}\boldsymbol{x}_{0|t} + \boldsymbol{n}_{0|t}$

8     $\boldsymbol{\Sigma}_t \leftarrow r_t^2 \boldsymbol{A}\boldsymbol{A}^\mathsf{T} + q_t^2 I$

9     $\boldsymbol{x}_t \leftarrow \boldsymbol{x}_t - \lambda r_t^2 (\nabla_{\boldsymbol{x}_t}\boldsymbol{x}_{0|t})\boldsymbol{A}^\mathsf{T}\boldsymbol{\Sigma}_t^{-1}(\boldsymbol{y} - \boldsymbol{\mu}_t)$

10    $\boldsymbol{n}_t \leftarrow \boldsymbol{n}_t - \kappa q_t^2 (\nabla_{\boldsymbol{n}_t}\boldsymbol{n}_{0|t}) \quad \boldsymbol{\Sigma}_t^{-1}(\boldsymbol{y} - \boldsymbol{\mu}_t)$

11   ...

12 ...

     // Unconditional diffusion steps

13     $\boldsymbol{x}_{t-\Delta t} \leftarrow \boldsymbol{x}_t - f(t)\boldsymbol{x}_t \Delta t$

14     $\boldsymbol{x}_{t-\Delta t} \leftarrow \boldsymbol{x}_{t-\Delta t} + g(t)^2 s_\theta^*(\boldsymbol{x}_t, t)\Delta t$

15     $\mathbf{z} \sim \mathcal{N}(\mathbf{0}, \mathbf{I})$

16     $\boldsymbol{x}_{t-\Delta t} \leftarrow \boldsymbol{x}_{t-\Delta t} + g(t)\sqrt{\Delta t}\mathbf{z}$

17

18     $\boldsymbol{n}_{t-\Delta t} \leftarrow \boldsymbol{n}_t - f(t)\boldsymbol{n}_t \Delta t$

19     $\boldsymbol{n}_{t-\Delta t} \leftarrow \boldsymbol{n}_{t-\Delta t} + g(t)^2 s_\phi^*(\boldsymbol{n}_t, t)\Delta t$

20     $\mathbf{z} \sim \mathcal{N}(\mathbf{0}, \mathbf{I})$

21     $\boldsymbol{n}_{t-\Delta t} \leftarrow \boldsymbol{n}_{t-\Delta t} + g(t)\sqrt{\Delta t}\mathbf{z}$

22 **end**

    **return:** $\boldsymbol{x}_0$

---

## 4   Related Work

In this section, we discuss alternative approaches for tackling inverse problems with structured noise using deep generative models, namely normalizing flows (NF) and generative adversarial networks (GAN). These methods, along with three widely used diffusion posterior sampling methods, serve as baselines in our experiments to evaluate the performance of the proposed diffusion-based denoiser. Importantly, while the diffusion methods included in the comparison do not explicitly model the noise prior, our approach is the first to tackle structured noise in inverse problem settings. Direct application of existing diffusion posterior sampling methods without the proposed joint-sampling framework fails to effectively remove structured noise, as shown in our experimental results. That being said, we do show the compatibility of our method with current state-of-the-art guided diffusion samplers. Finally, the NF- and GAN-based methods discussed in the following section rely on MAP estimation, see Section 2, whereas we perform posterior sampling.

**Normalizing Flows:** Whang et al. (2021) propose to use normalizing flows to model both the data and the noise distributions. Normalizing flows are a special class of likelihood-based generative models that make use of an invertible mapping $G : \mathbb{R}^d \to \mathbb{R}^d$ to transform samples from a base distribution $p_Z(\boldsymbol{z})$ into a more complex multimodal distribution $\boldsymbol{x} = G(\boldsymbol{z}) \sim p_X(\boldsymbol{x})$. The invertible nature of the mapping $G$ allows for exact density evaluation through the change of variables formula:

$$\log p_X(\boldsymbol{x}) = \log p_Z(\boldsymbol{z}) + \log|\det J_{G^{-1}}(\boldsymbol{x})|, \tag{16}$$

where $J$ is the Jacobian that accounts for the change in volume between densities. Since exact likelihood computation is possible through the flow direction $G^{-1}$, the parameters of the generator network can be optimized to maximize likelihood of the training data. Subsequently, the inverse task is solved using the MAP estimation in equation 3:

$$\hat{\boldsymbol{x}} = \arg\max_{\boldsymbol{x}} \{\log p_{G_N}(\boldsymbol{y} - \boldsymbol{A}\boldsymbol{x}) + \log p_{G_X}(\boldsymbol{x})\}, \tag{17}$$

where $G_N$ and $G_X$ are generative flow models for the noise and data respectively. Analog to that, the solution can be solved in the latent space rather than the image space as follows:

$$\hat{\boldsymbol{z}} = \arg\max_{\boldsymbol{z}}\left\{\log p_{G_N}(\boldsymbol{y} - \boldsymbol{A}(G_X(\boldsymbol{z}))) + \lambda \log p_{G_X}(G_X(\boldsymbol{z}))\right\}. \tag{18}$$

Note that in equation 18 a smoothing parameter $\lambda$ is added to weigh the prior and likelihood terms, as was also done in Whang et al. (2021). The optimal $\hat{\boldsymbol{x}}$ or $\hat{\boldsymbol{z}}$ can then be found by applying gradient ascent on equation 17 or equation 18, respectively.

**Generative Adversarial Networks:** Generative adversarial networks are implicit generative models that can learn the data manifold in an adversarial manner (Goodfellow et al., 2020). The generative model is trained with an auxiliary discriminator network that evaluates the generator's performance in a minimax game. The generator $G(\boldsymbol{z}) : \mathbb{R}^l \to \mathbb{R}^d$ maps latent vectors $\boldsymbol{z} \in \mathbb{R}^l \sim \mathcal{N}(\boldsymbol{0}, \boldsymbol{I})$ to the data distribution of interest. The structure of the generative model can also be used in inverse problem solving (Bora et al., 2017). The objective can be derived from equation 1 and is given by:

$$\hat{\boldsymbol{z}} = \arg\min_{\boldsymbol{z}}\left\{||\boldsymbol{y} - AG_X(\boldsymbol{z})|| + \lambda ||z||_2^2\right\}, \tag{19}$$

where $\lambda$ weights the importance of the prior with the measurement error. Similar to NF, the optimal $\hat{\boldsymbol{z}}$ can be found using gradient ascent. The $\ell_2$ regularization term on the latent variable is proportional to negative log-likelihood under the prior defined by $G_X$, where the subscript denotes the density that the generator is approximating. While this method does not explicitly model the noise, it remains an interesting comparison, as the generator cannot reproduce the noise found in the measurement and can only recover signals that are in the range of the generator. Therefore, due to the limited support of the learned distribution, GANs can inherently remove structured noise. However, the representation error (i.e. observation lies far from the range of the generator (Bora et al., 2017)) imposed by the structured noise comes at the cost of recovery quality.

## 5 Implementation Details

Automatic hyperparameter tuning for optimal inference was performed for the proposed and all baseline methods on a small validation set of only 5 images (depending on the experiment as detailed in section 6). All parameters used for training and inference can be found in the provided code repository linked in the paper. A summary of the most important hyperparameters for each method can be found in Appendix C. The peak signal-to noise ratio (PSNR), structural similarity index (SSIM) and perceptual similarity metric (LPIPS) (Zhang et al., 2018) are used to evaluate our results and inspect both ends of the *perception-distortion tradeoff* (Blau & Michaeli, 2018).

### 5.1 Proposed Method

Given the two separate datasets, one for the data and one for the structured noise, two separate score models can be trained independently. This allows for easy adaptation of our method, since many existing trained score models can be reused. Furthermore, this ensures the same two prior networks can be used in a variety of different tasks. For both the score models, we use the NCSNv2 architecture as introduced in Song & Ermon (2020). The two priors are combined only during inference through the proposed sampling procedure as described in Algorithm 1, using the adapted Euler-Maruyama sampler. We use the following SDE: $f(t) = 0$, $g(t) = \sigma^t$ with $\sigma = 25$ to define the diffusion trajectory. During each experiment, we run the sampler for $T = 600$ iterations.

### 5.2 Baseline Methods

As a starting point, we compare our method across all experiments with three common diffusion posterior sampling approaches. Unlike our proposed framework, these methods rely on a Gaussian noise prior and do not utilize an explicitly learned noise model.

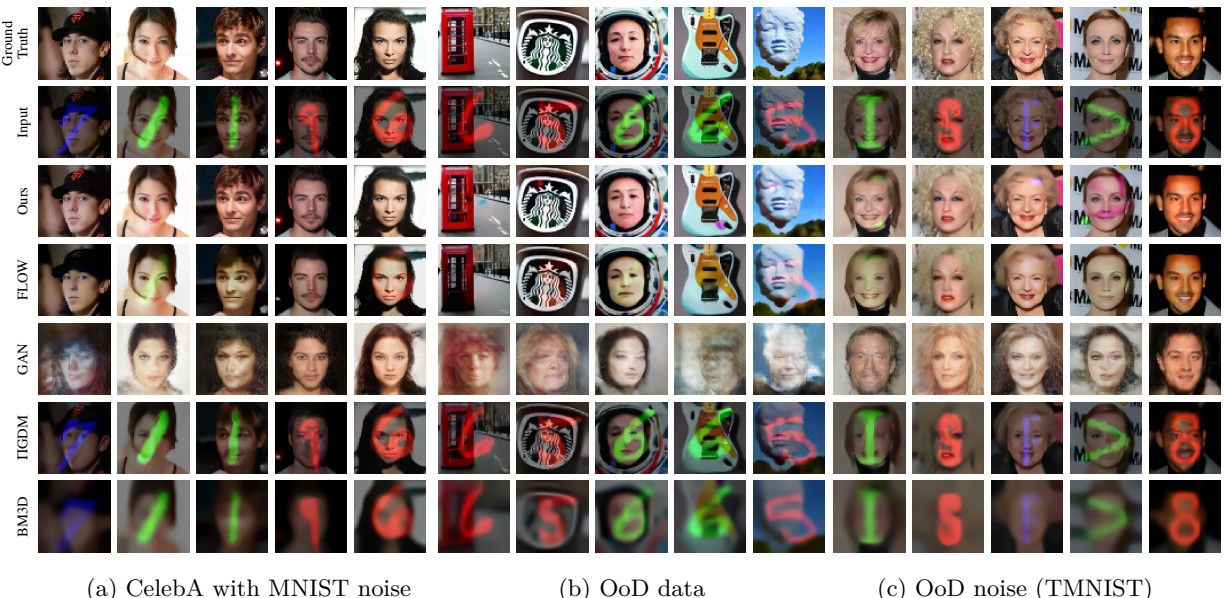

   (a) CelebA with MNIST noise      (b) OoD data      (c) OoD noise (TMNIST)

Figure 2: Qualitative results on the removing MNIST digits (noise) from CelebA (signal) experiment, comparing our joint posterior sampling method to the baselines:[2]: ◇ΠGDM, †FLOW, ‡GAN, §BM3D.

The closest to our work is the flow-based noise model proposed by Whang et al. (2021), discussed in Section 4, which will serve as our main baseline. To boost the performance of this baseline and to make it more competitive, we moreover replace the originally used RealNVP (Dinh et al., 2016) with the Glow architecture (Kingma & Dhariwal, 2018). We use the exact implementation found in Asim et al. (2020), with a flow depth of $K = 18$, and number of levels $L = 4$, which has been optimized for the same CelebA dataset used in this work and thus should provide a fair comparison with the proposed method.

Additionally, GANs, as discussed in Section 4, are used as a comparison. We train a DCGAN (Radford et al., 2015), with a generator latent input dimension of $l = 100$. The generator architecture consists of 4 strided 2D transposed convolutional layers, having $4 \times 4$ kernels yielding feature maps of 512, 256, 128 and 64. Each convolutional layer is followed by a batch normalization layer and ReLU activation.

Lastly, depending on the reconstruction task, classical non-data-driven methods are used as a comparison. For denoising experiments, we use the block-matching and 3D filtering algorithm (BM3D) (Dabov et al., 2006), and in compressed sensing experiments, LASSO with wavelet basis (Tibshirani, 1996). Except for the flow-based method of Whang et al. (2021), none of these methods explicitly model the noise distribution. Still, they are a valuable baseline, as they demonstrate the effectiveness of incorporating a learned structured noise prior rather than relying on simple noise priors.

## 6  Experiments

We subject our method to a variety of inverse problems such as denoising, compressed sensing, deraining, and dehazing, all with an element of additive structured noise. To test the method's robustness, we repeat the experiments on both *out-of-distribution* (OoD) data and OoD noise in Section 6.2. To show the capabilities of our method in a variety of contexts, we evaluate the joint-conditional diffusion method on different datasets, such as CelebA (Section 6.1, 6.2, 6.3), ImageNet (Section 6.2), FFHQ (Section 6.4), and a medical ultrasound dataset (Section 6.5). Lastly, we compare the methods' computational performance in Section 6.6. The proposed method outperforms the baselines both qualitatively and quantitatively in all experiments.

### 6.1 Removing MNIST digits from CelebA

**Setup:** For comparison with Whang et al. (2021), we recreate an experiment introduced in their work, where MNIST digits are added to CelebA faces. The corruption process is defined by $\boldsymbol{y} = 0.5 \cdot \boldsymbol{x}_{\mathrm{CelebA}} + 0.5 \cdot \boldsymbol{n}_{\mathrm{MNIST}}$. In the experiment, the signal score network $s_\theta$ is trained on the CelebA dataset (Liu et al., 2015) and the noise score network $s_\phi$ on the MNIST dataset, with 10000 and 27000 training samples, respectively. Images are resized to $64 \times 64$ pixels. We test on a randomly selected subset of 100 images.

**Results:** A random selection of test samples is shown in Fig. 2a for qualitative analysis. Additionally, Fig. 3a shows a quantitative comparison of our method against all baselines. Both our proposed diffusion method and the flow-based method have a an explicit noise prior and are able to recover the underlying signal, with the diffusion method preserving more details. While the GAN method effectively removes the digits, it struggles to accurately reconstruct the faces, as it fails to project the observations onto the range of the generator. Both the BM3D denoiser as well as the diffusion method without structured noise prior (ΠGDM) fail to recover the underlying signal, confirming the importance of prior knowledge of the noise.

### 6.2 Out-of-distribution data and noise

**Setup:** In real-world applications, both signal and noise are often subject to distribution shifts with respect to the original training data, making it challenging to train reliable models. While in many practical cases both signal and noise components can be measured in isolation or simulated, the resulting data may not perfectly match the true underlying distributions. This motivates the need to evaluate the robustness of models under out-of-distribution (OoD) conditions.

To this end, we extend our previous experiments in Section 6.1 to include OoD scenarios for both signals and noise. Specifically for the signal case, we test with (1) random data from ImageNet and (2) synthetically generated data from the Stable Diffusion text-to-image model (Rombach et al., 2022). To explore the robustness to shift in noise distribution, we introduce two OoD noise variants: (1) samples drawn from the TMNIST-Alphabet dataset, which features different characters, and (2) random translations applied to the noise (digits). Importantly, we use the exact same hyperparameters and models as in the original non-OoD experiments.

**Results:** Qualitative results for the OoD data and noise experiments are shown in Fig.2b and Fig.2c, respectively. Consistent with prior findings (Asim et al., 2020; Whang et al., 2021), the flow-based method shows robustness to OoD data, unlike the GAN. We empirically show that the diffusion method is also resistant to OoD data and noise in inverse tasks with complex noise structures and demonstrates superior performance over the baselines. Quantitative results for the OoD data experiment are shown in Fig.3b, while we refer the reader to Appendix B.1 for extended results on the OoD noise experiments. Among the OoD noise variants, random translations proved more challenging than TMNIST-Alphabet characters, with our method maintaining its competitive edge.

### 6.3 Compressed sensing with structured noise

**Setup:** In this experiment, the corruption process is defined by $\boldsymbol{y} = \boldsymbol{A}\boldsymbol{x} + \boldsymbol{n}_{\mathrm{sine}}$ with a random Gaussian measurement matrix $\boldsymbol{A} \in \mathbb{R}^{m \times d}$ and a noise source with sinusoidal variance $\sigma_k \propto \exp(\sin(\frac{2\pi k}{16}))$ for each pixel $k$, which we use to train $s_\phi$. The subsampling factor is defined by the size of the measurement matrix $d/m$. Additionally, we include an experiment with the special case $\boldsymbol{A} = \boldsymbol{I}$ and a 2D sinusoidal noise pattern, where $k$ is now each row in the image.

**Results:** In Fig. 4a the results of the compressed sensing experiment and the comparison with the baselines are shown for an average standard deviation of $\sigma_N = 0.2$ and subsampling of factor $d/m = 2$. The proposed method demonstrates robust recovery under structured noise and distribution shifts in out-of-distribution (OoD) cases. In contrast, the flow-based method underperforms when subjected to the OoD data, see Fig. 4b. A qualitative analysis is shown in Appendix B.2. Interestingly, DPS (diffusion without explicit noise model)

---

[2]⋆Ours, ⋄(Song et al., 2023), †(Whang et al., 2021), ‡(Bora et al., 2017), §(Dabov et al., 2006), ¶(Tibshirani, 1996)

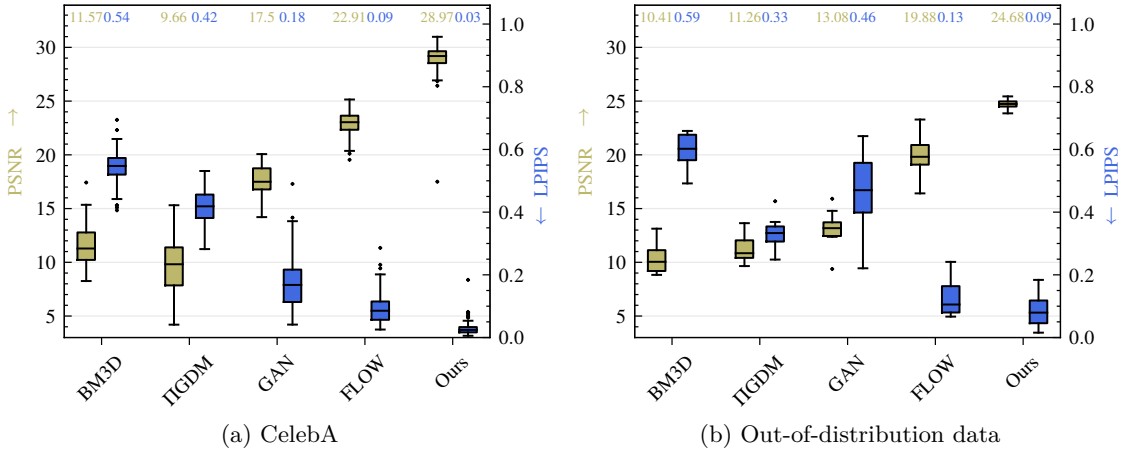

(a) CelebA

(b) Out-of-distribution data

Figure 3: Quantitative results using PSNR (green) and LPIPS (blue) for the removing MNIST digits experiment of the (a) CelebA and (b) out-of-distribution datasets.

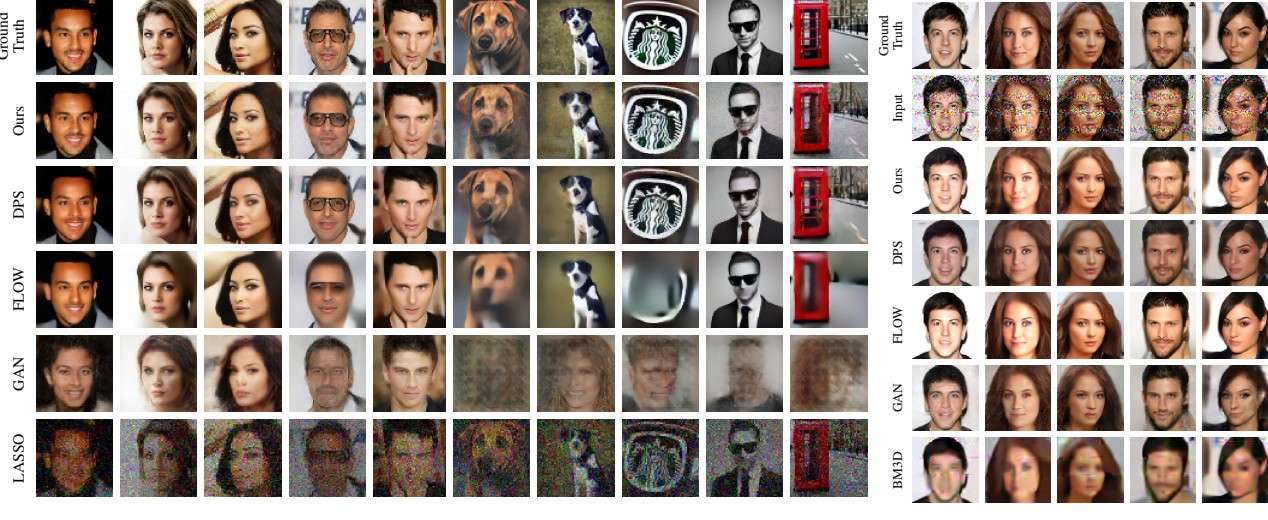

(a) CelebA with structured noise      (b) Out-of-distribution data

Figure 4: Results on the compressed sensing with structured noise experiment, comparing our diffusion-based method to the baselines.

Figure 5: Results on the 2D sinusoidal noise experiment.

performs relatively well in this CS experiment, which is likely due to the random mapping of the Gaussian noise pattern through the measurement matrix reduces the structure of the noise. The necessity of a learned noise prior becomes more apparent in Fig. 5, where DPS is unable to obtain an accurate estimate of the signal. For detailed results see Appendix B.2.

## 6.4 Deraining FFHQ

**Setup:** In this experiment, we address the problem of *deraining*, which involves removing rain streaks from images significantly occluding objects of interest. We employ the $256 \times 256$ FFHQ dataset to assess our method's performance on high-resolution images. In this setup, the signal diffusion model is trained on the FFHQ dataset, whereas the noise model is trained using a rain simulator.

**Results:** We compare our method to diffusion posterior sampling without an explicit noise model in Fig. 7. The proposed method scores $\text{PSNR} = 23.97, \text{SSIM} = 0.82, \text{LPIPS} = 0.18$. Unsurprisingly, we observe that

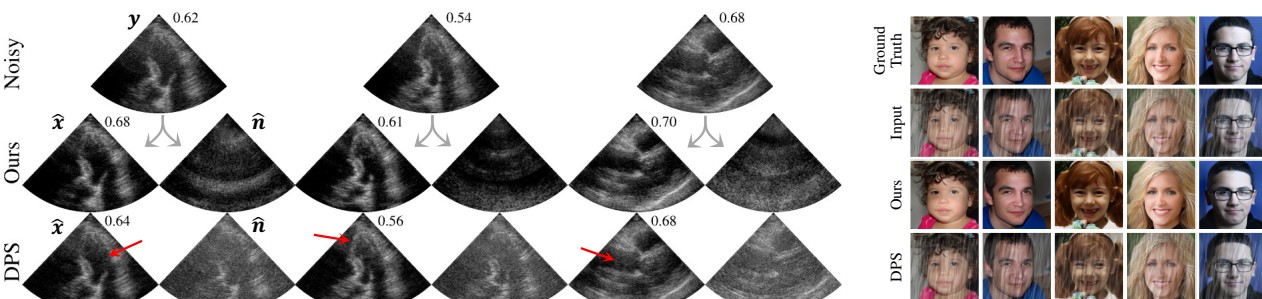

Figure 6: Comparison of diffusion posterior sampling methods with explicit noise model (Ours) and without (DPS) on the task of dehazing ultrasound data. The gCNR metric is given for each example.

Figure 7: Deraining experiment on the FFHQ $256 \times 256$ dataset comparing DPS with ours.

diffusion models without explicit modeling of the noise distribution (DPS) fail to accurately reconstruct the images under heavy structured noise, as it scores $\mathrm{PSNR} = 19.70, \mathrm{SSIM} = 0.67, \mathrm{LPIPS} = 0.40$.

## 6.5 Medical Ultrasound Reconstruction

**Setup:** To evaluate the proposed method in a realistic medical imaging context, we address the inverse problem of *dehazing* in cardiac ultrasound, aiming to reconstruct a clear depiction of the anatomy from hazy observations. The haze artifact arises from multipath scattering between the probe and tissue of interest. We train our diffusion model on log-compressed beamformed IQ data, and model the observed data as $\boldsymbol{y} = \boldsymbol{x} + \boldsymbol{n}$, where $\boldsymbol{x}$ represents signals originating from tissue and $\boldsymbol{n}$ corresponds to the multipath scattering. The signal dataset is constructed using clean images minimally impacted by haze, while the noise dataset is acquired by capturing data from an ultrasound probe scanning a medium with high scattering.

**Results:** We evaluate the method on a cardiac ultrasound dataset, acquired with Philips X51-c probe, using the unsupervised generalized contrast-to-noise ratio (gCNR ↑) metric (Rodriguez-Molares et al., 2020), yielding values of 0.58, 0.62, and 0.65 for the noisy input, DPS, and the proposed method, respectively, across a test set of 100 images. Fig. 6 presents qualitative results, including noise estimates from both methods. Unlike the proposed method, DPS leaves residual signal in its noise estimate, effectively "eating away" at the clean signal, which is often unacceptable in a clinical context. In contrast, the proposed method produces noise estimates that closely resemble the haze, effectively suppressing the hazy regions without distorting the underlying anatomy, resulting in clearer images.

## 6.6 Performance

To highlight the difference in inference time between our method and the baselines, benchmarks are performed on a single 12GBytes NVIDIA GeForce RTX 3080 Ti, see Table 5 in Appendix B. A quick comparison of inference times reveals a $4\times$ ($\Pi$GDM) or $10\times$ (Projection) difference in speed between ours and the flow-based method. All the deep generative models need approximately an equal amount of iterations ($T \approx 600$) to converge. However, given the same modeling capacity, the flow model requires substantially more trainable parameters compared to the diffusion method. This is mainly due to the restrictive requirements imposed on the architecture to ensure tractable likelihood computation. It should be noted that in this work no improvements are applied to speed up the diffusion process, such as distillation (Salimans & Ho, 2021) or improved initialization (Chung et al., 2022c), leaving room for even more improvement in future work.

## 7 Discussions

Inverse problems are powerful tools for inferring unknown signals from observed measurements and have been at the center of many signal and image processing algorithms. Strong priors, often those learned through deep generative models, have played a crucial role in guiding these inferences, especially in the context of

high-dimensional data. While complex priors on the signal are commonly employed, noise sources are often assumed to be simply distributed, drastically reducing their effectiveness in structured noise settings.

In this work, we address this limitation by introducing a novel joint posterior sampling technique. We not only leverage deep generative models to learn strong priors for the signal, but we also extend our approach to incorporate priors on the noise distribution. To achieve this, we employ an additional diffusion model that has been trained specifically to capture the characteristics of structured noise. Furthermore, we show the compatibility of our method with three existing posterior sampling techniques (projection, DPS, ΠGDM). We demonstrate our method on natural and out-of-distribution data and noise and achieve increased performance over the state-of-the-art and established conventional methods for complex inverse tasks. Additionally, the diffusion-based method is substantially easier to train using the score matching objective compared to other deep generative methods that rely on constrained neural architectures or adversarial training.

While our method shows considerable improvements in speed and effectiveness at removing structured noise compared to the flow-based method, it is not yet suitable for real-time inference and still lags behind GANs and classical methods in terms of inference speed. Fortunately, research into accelerating the diffusion process is on its way. In addition, although a simple sampling algorithm was adopted in this work, many more sampling algorithms for score-based diffusion models exist. Future work should explore this wide increase in design space to understand the limitations and possibilities of more sophisticated sampling schemes in combination with the proposed joint posterior sampling method. Additionally, our method assumes independent noise and linear measurement models. Extending to a broader family of possibly non-linear or dependent cases is an interesting direction for future work. Lastly, the connection between diffusion models and continuous normalizing flows through the neural ODE formulation (Song et al., 2021a) is not investigated but is of great interest given the comparison with the flow-based method in this work.

## 8   Conclusions

In this work, we presented a framework for removing structured noise using diffusion models. The proposed joint posterior sampling technique for diffusion models has been shown to effectively remove highly structured noise and outperform baselines in both image quality and computational performance. Additionally, it exhibits enhanced robustness in out-of-distribution scenarios. Our work provides an efficient addition to existing score-based conditional sampling methods by incorporating knowledge of the noise distribution, whilst supporting a variety of guided diffusion samplers. Future work should focus on accelerating the relatively slow inference process of diffusion models and further investigate the applicability of the proposed method outside the realm of natural images.

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

# A  Derivation of Data Consistency Steps

The proposed joint posterior sampling framework for removing structured noise is versatile and compatible with various existing diffusion posterior sampling methods. In Section 3.2 we established the foundation for jointly sampling from two distributions (signal and noise), given a corrupted observation. Specifically, a derivation was given for the ΠGDM data consistency method. The following section presents two additional examples of leveraging popular guidance methods for diffusion models to remove structured noise, namely DPS and projection. A comparison of all methods is shown in Appendix B.4.

## A.1  DPS

Diffusion Posterior Sampling (DPS) (Chung et al., 2022a) also leverages Tweedie's formula in order to estimate $\boldsymbol{x}_{0|t}$ and $\boldsymbol{n}_{0|t}$. However, unlike ΠGDM, DPS does not leverage VI with Gaussian posteriors. Instead, a Gaussian error with diagonal covariance and variance $\rho^2$ is assumed, which again we can adapt to our problem as such:

$$p(\boldsymbol{y}|\boldsymbol{x}_t, \boldsymbol{n}_t) \approx \mathcal{N}(\boldsymbol{\gamma}_t; \boldsymbol{\mu}_t, \boldsymbol{\Sigma}_t) \begin{cases} \boldsymbol{\gamma}_t = \boldsymbol{y} \\ \boldsymbol{\mu}_t = \boldsymbol{A}\boldsymbol{x}_{0|t} + \boldsymbol{n}_{0|t} \\ \boldsymbol{\Sigma}_t = \rho^2 I, \end{cases} \tag{20}$$

resulting in the following scores:

$$\begin{bmatrix} \nabla_{\boldsymbol{x}_t} \log p(\boldsymbol{y}|\boldsymbol{x}_t, \boldsymbol{n}_t) \\ \nabla_{\boldsymbol{n}_t} \log p(\boldsymbol{y}|\boldsymbol{x}_t, \boldsymbol{n}_t), \end{bmatrix} \approx \begin{bmatrix} \frac{1}{\rho^2}(\nabla_{\boldsymbol{x}_t} \boldsymbol{x}_{0|t}) \ \boldsymbol{A}^\mathsf{T}(\boldsymbol{y} - A\boldsymbol{x}_{0|t} - \boldsymbol{n}_{0|t}) \\ \frac{1}{\rho^2}(\nabla_{\boldsymbol{n}_t} \boldsymbol{n}_{0|t}) \quad (\boldsymbol{y} - A\boldsymbol{x}_{0|t} - \boldsymbol{n}_{0|t}) \end{bmatrix}, \tag{21}$$

Note the difference between equations (15) and (21). The former employs a non-diagonal covariance matrix, while the latter uses a simple diagonal approximation. In other words, DPS does not take into account how the variance of the estimation of $\boldsymbol{x}_{0|t}$ gets mapped to $\boldsymbol{y}$, in the case of a non-diagonal measurement matrix $\boldsymbol{A}$. The authors of DPS (Chung et al., 2022a) then propose to rescale the noise, or step size, of the noise-perturbed likelihood score by a fixed scalar divided by the norm of the noise-perturbed likelihood. Additionally, the diffusion coefficient $g(t)^2$ gets canceled out in the weighting scheme. Again, we achieve that here by choosing $\lambda$ and $\kappa$ appropriately.

## A.2  Projection

The projection method (Song et al., 2020) takes another approach altogether in comparison with ΠGDM and DPS. Instead of relating $\boldsymbol{x}_t, \boldsymbol{n}_t$ toward $\boldsymbol{x}_0, \boldsymbol{n}_0$, it relates $\boldsymbol{y}$ to $\boldsymbol{y}_t$ and then uses the following approximation:

$$p(\boldsymbol{y}|\boldsymbol{x}_t, \boldsymbol{n}_t) \approx p(\hat{\boldsymbol{y}}_t|\boldsymbol{x}_t, \boldsymbol{n}_t), \tag{22}$$

where $\hat{\boldsymbol{y}}_t$ is a sample from $p(\boldsymbol{y}_t|\boldsymbol{y})$, and $\{\boldsymbol{y}_t\}_{t \in [0,1]}$ is an additional stochastic process that essentially corrupts the observation along the SDE trajectory together with $\boldsymbol{x}_t$. Note that in the case of a linear measurement $p(\boldsymbol{y}_t|\boldsymbol{y})$ is tractable, and we can easily compute $\hat{\boldsymbol{y}}_t = \alpha_t \boldsymbol{y} + \beta_t \boldsymbol{A}\boldsymbol{z}$, using the reparameterization trick with $\boldsymbol{z} \in \mathbb{R}^d \sim \mathcal{N}(\boldsymbol{0}, \boldsymbol{I})$, see a follow-up paper of the same group; Song et al. (2021b). In contrast to the case where we use DPS and ΠGDM, which perform the data consistency using noiseless estimates at diffusion time $t = 0$, the projection method projects the observation to the current diffusion step $t$. Consequently, we cannot sample the noise vectors $\boldsymbol{z}$ independently anymore, but should reuse them for the forward diffusion of signal, noise and observation.

We then use the measurement model which is normally only defined for time $t = 0$, and apply it to the current timestep $t$. In this approximation, we assume that we make a Gaussian error with diagonal covariance and standard deviation $\rho^2$ as:

$$p(\boldsymbol{y}|\boldsymbol{x}_t, \boldsymbol{n}_t) \approx \mathcal{N}(\boldsymbol{\gamma}_t; \boldsymbol{\mu}_t, \boldsymbol{\Sigma}_t) \begin{cases} \boldsymbol{\gamma}_t = \hat{\boldsymbol{y}}_t \\ \boldsymbol{\mu}_t = \boldsymbol{A}\boldsymbol{x}_t + \boldsymbol{n}_t \\ \boldsymbol{\Sigma}_t = \rho^2 I. \end{cases} \tag{23}$$

Calculating the score of equation 23 with respect to both $\boldsymbol{x}_t$ and $\boldsymbol{n}_t$ then results in:

$$\begin{bmatrix} \nabla_{\boldsymbol{x}_t} \log p(\boldsymbol{y}|\boldsymbol{x}_t, \boldsymbol{n}_t) \\ \nabla_{\boldsymbol{n}_t} \log p(\boldsymbol{y}|\boldsymbol{x}_t, \boldsymbol{n}_t) \end{bmatrix} \approx \begin{bmatrix} \frac{1}{\rho^2} \boldsymbol{A}^{\mathsf{T}} (\hat{\boldsymbol{y}}_t - \boldsymbol{A}\boldsymbol{x}_t - \boldsymbol{n}_t) \\ \frac{1}{\rho^2} \quad (\hat{\boldsymbol{y}}_t - \boldsymbol{A}\boldsymbol{x}_t - \boldsymbol{n}_t) \end{bmatrix}, \tag{24}$$

Similar to DPS, we reweigh the scores in order to cancel out both $g(t)^2$ and $1/\rho^2$, using $\lambda$ and $\kappa$, see Table 1.

## B  Extended results

In the following section, we complement the experiments outlined in Section 6 with further analysis. Additionally, there are other experiments, such as a comparison of data consistency methods used in combination with the proposed joint-sampling framework; see Appendix B.4.

### B.1  Out-of-distribution data and noise

Table 2: Results for the OoD signal and noise (TMNIST or translation) experiments in Section 6.2. **Problem:** $\boldsymbol{y} = 0.5 \cdot \boldsymbol{x}_{\text{CelebA} \vee \text{OoD}} + 0.5 \cdot \boldsymbol{n}_{\text{TMNIST} \vee \text{translation}}$.

|  | Dataset | TMNIST | | | translation | | |
|---|---|---|---|---|---|---|---|
|  |  | PSNR (↑) | SSIM (↑) | LPIPS (↓) | PSNR (↑) | SSIM (↑) | LPIPS (↓) |
| Noisy | CelebA | $12.26 \pm 2.0$ | $0.633 \pm 0.02$ | $0.305 \pm 0.11$ | $12.14 \pm 2.0$ | $0.634 \pm 0.02$ | $0.279 \pm 0.07$ |
| *Ours | CelebA | $25.94 \pm 2.4$ | $0.851 \pm 0.04$ | $0.159 \pm 0.06$ | $23.63 \pm 4.1$ | $0.893 \pm 0.04$ | $0.150 \pm 0.07$ |
| †FLOW | CelebA | $22.61 \pm 1.1$ | $0.826 \pm 0.05$ | $0.179 \pm 0.06$ | $22.96 \pm 1.1$ | $0.837 \pm 0.05$ | $0.167 \pm 0.05$ |
| Noisy | OoD | $11.50 \pm 1.3$ | $0.634 \pm 0.01$ | $0.251 \pm 0.08$ | $11.49 \pm 1.3$ | $0.641 \pm 0.01$ | $0.203 \pm 0.08$ |
| *Ours | OoD | $22.59 \pm 2.4$ | $0.858 \pm 0.06$ | $0.197 \pm 0.08$ | $21.55 \pm 3.0$ | $0.895 \pm 0.05$ | $0.167 \pm 0.09$ |
| †FLOW | OoD | $20.06 \pm 1.8$ | $0.831 \pm 0.08$ | $0.177 \pm 0.07$ | $20.54 \pm 2.1$ | $0.839 \pm 0.08$ | $0.157 \pm 0.06$ |

### B.2  Compressed sensing

Table 3: Quantitative results for compressed sensing experiments as outlined in Section 6.3. **Problem:** $\boldsymbol{y} = \boldsymbol{A}\boldsymbol{x} + \boldsymbol{n}_{\text{sine}}$, $\boldsymbol{A} \in \mathbb{R}^{m \times d}$, $d/m = 2$, $\boldsymbol{n}_{\text{sine}} \sim \mathcal{N}(0, \sigma_k^2)$, $\sigma_k \propto \exp(\sin(\frac{2\pi k}{16}))$ for each pixel $k$.

|  | CelebA | | | OoD | | |
|---|---|---|---|---|---|---|
|  | PSNR (↑) | SSIM (↑) | LPIPS (↓) | PSNR (↑) | SSIM (↑) | LPIPS (↓) |
| *Ours | $25.51 \pm 1.0$ | $0.823 \pm 0.04$ | $0.042 \pm 0.02$ | $22.90 \pm 1.6$ | $0.823 \pm 0.08$ | $0.059 \pm 0.02$ |
| •DPS | $25.34 \pm 1.0$ | $0.820 \pm 0.05$ | $0.052 \pm 0.03$ | $22.79 \pm 1.6$ | $0.818 \pm 0.09$ | $0.069 \pm 0.04$ |
| †FLOW | $24.96 \pm 2.3$ | $0.779 \pm 0.08$ | $0.105 \pm 0.07$ | $19.85 \pm 4.8$ | $0.608 \pm 0.18$ | $0.266 \pm 0.16$ |
| ‡GAN | $18.90 \pm 1.3$ | $0.529 \pm 0.08$ | $0.136 \pm 0.06$ | $12.39 \pm 1.7$ | $0.159 \pm 0.07$ | $0.518 \pm 0.14$ |
| ¶LASSO | $12.93 \pm 1.8$ | $0.284 \pm 0.04$ | $0.645 \pm 0.08$ | $11.62 \pm 1.5$ | $0.336 \pm 0.06$ | $0.493 \pm 0.10$ |

Table 4: Quantitative results for the structured sinusoidal noise experiments as outlined in Section 6.3. **Problem:** $\boldsymbol{y} = \boldsymbol{x} + \boldsymbol{n}_{\text{sine}}$, $\boldsymbol{n}_{\text{sine}} \sim \mathcal{N}(0, \sigma_k^2)$, $\sigma_k \propto \exp(\sin(\frac{2\pi k}{16}))$ for each row $k$.

|  | CelebA | | |
|---|---|---|---|
|  | PSNR (↑) | SSIM (↑) | LPIPS (↓) |
| Noisy | $15.60 \pm 0.3$ | $0.434 \pm 0.06$ | $0.458 \pm 0.11$ |
| *Ours | $18.61 \pm 1.0$ | $0.772 \pm 0.05$ | $0.054 \pm 0.02$ |
| •DPS | $22.31 \pm 1.0$ | $0.716 \pm 0.06$ | $0.074 \pm 0.04$ |
| †FLOW | $18.11 \pm 1.7$ | $0.800 \pm 0.05$ | $0.070 \pm 0.04$ |
| ‡GAN | $21.15 \pm 1.5$ | $0.632 \pm 0.07$ | $0.097 \pm 0.04$ |
| §BM3D | $23.12 \pm 1.0$ | $0.695 \pm 0.06$ | $0.209 \pm 0.06$ |

---

[2]*Ours, ◇(Song et al., 2023), •(Chung et al., 2022a), †(Whang et al., 2021), ‡(Bora et al., 2017), §(Dabov et al., 2006), ¶(Tibshirani, 1996)

### B.3 Performance

A summary of the performance of the proposed methods and baselines as discussed in Section 6.6 is listed in Table 5.

Table 5: Inference performance benchmark for all methods.

| Model | | # trainable parameters | Inference time [ms] |
|---|---|---|---|
| [*]Ours | (Proj.) | 8.9M | 5605 |
| | (DPS) | | 16818 |
| | (ΠGDM) | | 16094 |
| [†]FLOW | | 25.8M | 61853 |
| [‡]GAN | | 3.9M | 59 |
| [§]BM3D | | – | 29 |

### B.4 Comparison Data Consistency Methods

The proposed joint sampling framework outperforms any of the baselines mentioned in Section 4, regardless of which of the three diffusion-based data consistency methods are used as basis, see Section 3.2 (ΠGDM), and Appendix A (DPS, projection). Nonetheless, we investigate how the specific data-consistency rule used affects the performance of our method in the task of removing structured noise. As shown in Fig. 8a, ΠGDM as basis for our framework provides the most consistent results with lower variance between samples. Empirically, this trend continues to be seen in the out-of-distribution datasets; see Fig. 8b. This is not surprising as ΠGDM has a more sophisticated approximation for the noise-perturbed likelihood score compared to DPS and the projection method. A visual comparison is shown in Fig. 8c. Note that in all these experiments the samplers are used in combination with our proposed joint sampling framework. Straightforward inference without a learned model for the noise distribution is unable to effectively remove structured noise as seen in Fig 7.

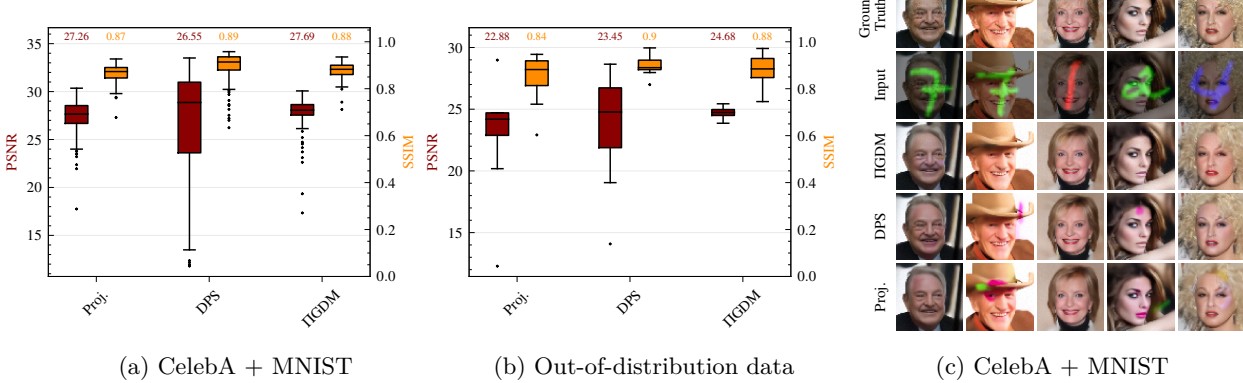

(a) CelebA + MNIST      (b) Out-of-distribution data      (c) CelebA + MNIST

Figure 8: Comparison of the proposed joint posterior sampling framework with different data consistency methods as basis (projection, DPS and ΠGDM). Qualitative (c) and quantitative results are shown using PSNR (red) and SSIM (orange) for the removing MNIST digits experiment on images of the (a) CelebA and (b) out-of-distribution datasets.

## C  Hyperparameters

For an extensive list of all hyperparameters used, consider looking at the configuration files for each experiment in the online codebase. A more compact summary can be found in Table 6.

Table 6: Hyperparameters for training and inference (CelebA + MNIST and related OoD experiments).

| Hyperparameters | | Diffusion | | | Flow | GAN |
|---|---|---|---|---|---|---|
| **Architecture** | | NCSNv2 VE-SDE $f(t) = 0,\ g(t) = \sigma^t = 25^t$ $\alpha_t = 1$ $\beta_t = \frac{1}{2\log\sigma}(\sigma^{2t} - 1)$ | | | Glow $L = 4$ (levels) $K = 18$ (depth) $c = 5$ (gradient clip norm) | DCGAN $l = 100$ (latent dim size) $4 \times 4$ kernel size $[512, 256, 128, 64]$ (channels generator) $[64, 128, 256, 512]$ (channels discriminator) |
| **Training** lr Adam epochs | | 0.0005 $\beta_1 = 0.9, \beta_2 = 0.999$ 150 | | | 0.0001 $\beta_1 = 0.9, \beta_2 = 0.999$ 300 | 0.0002 $\beta_1 = 0.5, \beta_2 = 0.999$ 100 |
| **Inference** DC rule | | **ΠGDM** | **DPS** | **proj.** | **gradient ascent (MAP)** | **gradient ascent (MAP)** |
| step size | | $1/T$ | $1/T$ | $1/T$ | 0.005 | 0.05 |
| $T$ | | 600 | 600 | 600 | 600 | 600 |
| $\lambda$ | | 0.93 | 12.7 | 0.5 | 0.03 | 0.9 |
| $\kappa$ | | 0.88 | 16.7 | 0.5 | - | - |
| $r_t^2$ | | $\beta_t^2/(\beta_t^2 - 1)$ | - | - | - | - |
| $q_t^2$ | | $\beta_t^2/(\beta_t^2 - 1)$ | - | - | - | - |

