# OpenReview forum: "Removing Structured Noise using Diffusion Models"
_TMLR — Accepted by TMLR_

### Review · Reviewer_DXv6 · 2024-11-29

**Summary Of Contributions:**

The authors propose a framework for removing structured noise using diffusion models, extending the posterior sampling approach to inverse problems with non-Gaussian noise models.

They do so by introducing a joint posterior sampling technique, utilizing an additional diffusion model which is trained to capture the characteristics of the structured noise.

They apply their method to inverse problems with synthetic structured noise, mainly based on the CelebA and MNIST datasets (e.g., adding MNIST digits to CelebA images and then recovering the clean CelebA images).

They compare their method mainly with normalizing flow and GAN models.

**Audience:**

Yes

**Broader Impact Concerns:**

No concerns.

**Claims And Evidence:**

No

**Requested Changes:**

The studied problem is interesting. This could definitely be relevant for the TMLR audience.

However, I have some concerns regarding the experimental evaluation, and the real-world applicability of the proposed methods requires some clarifications, se "Weaknesses" above.

**Strengths And Weaknesses:**

Strengths:
- The studied problem is interesting and relevant, extending the diffusion-based posterior sampling approach to structured noise models definitely seems like a good contribution to the field.
- The problem and proposed approach are introduced well in the Abstract, Introduction and Figure 1.
- The proposed method seems to work quite well on the current experiments.




Weaknesses:
- The paper could be more well-written overall. It contains some typos etc, and the methods and experiments/results are somewhat difficult to follow in Section 3 and 6, respectively.
- The proposed method is only applied to inverse problems with _synthetic_ structured noise. Even in the "Deraining FFHQ" experiment, synthetic rain streaks are added to images, and the noise model is trained on the same(?) type of simulated rain. It is unclear to me how the proposed method could be applied to real-world data with actual rain, fog or haze. Therefore, it is also unclear to me how useful this method would be in practice.




Other questions/suggestions:
- In eq. (8) you give separate expressions for $\nabla_{x_t}$ and $\nabla_{n_t}$, but in eq. (7) you should have $\nabla_{x_t, n_t}$?
- Why PSNR and SSIM as the only evaluation metrics, would it not be relevant to also include perceptual metrics such as LPIPS and FID?
- Is there really no diffusion-based method that could be relevant to include as a baseline? For example, what is the key difference between the proposed method and "Image Restoration with Mean-Reverting Stochastic Differential Equations" (Luo et al., ICML 2023)?
- Why are there no quantitative results reported for the experiments in Figure 4 or Figure 5?




Minor things:
- The formatting of Figure 4 and 5 is perhaps not ideal, maybe combine into sub-figures of a single figure?
- Figure 1 is never mentioned in the text?
- Eq. (1) is missing a "="?
- Should it not be "+" instead of "-" in eq. (3)?
- I think it could be helpful to add a short paragraph to the start of Section 3, clarifying what will be covered in the following subsections.
- Inconsistent use of e.g. "in equation 7" and "in (9)" (Table 1 caption) when referencing equations. Also, "Substitution of the VI estimate (12) into equation (11)" in Section 3.2.1.
- "Song et al. provide recommendations for choosing the variance of the VI Song et al. (2023)" in Section 3.2.1, odd citation formatting.
- Section 4, "Whang et al. propose to use normalizing flows to model both the data and the noise distributions Whang et al. (2021)", also odd citation formatting.
- Multiple "(Song et al., 2020)" citations should actually be "(Song et al., 2021b)", for the projection method, or?
- Figure 2 caption, "to the baselines2: †FLOW, ‡GAN, and §BM3D", but there are no footnotes anywhere?

---

> ### Author Response · Authors · 2025-01-14
> **Part 1**
>
> First of all we would like to thank the reviewer for their insightful comments and time dedicated to review our work.
>
> In the following we address the comments of the reviewer.
>
> > _The paper could be more well-written overall. It contains some typos etc, and the methods and experiments/results are somewhat difficult to follow in Section 3 and 6, respectively._
>
> We apologies for the several typos and have thoroughly checked the manuscript to clean those up. Additionally, we have rewritten Section 3 and 6 based on the reviewer's suggestion. Lastly, we have moved some of the experiments to the Appendix for better flow of the main body.
>
> > _The proposed method is only applied to inverse problems with _synthetic_ structured noise. Even in the "Deraining FFHQ" experiment, synthetic rain streaks are added to images, and the noise model is trained on the same(?) type of simulated rain. It is unclear to me how the proposed method could be applied to real-world data with actual rain, fog or haze. Therefore, it is also unclear to me how useful this method would be in practice._
>
> We appreciate the concerns about real-world applicability. Our method requires separate samples of each distribution ($x$ and $n$). The benefit of this method is that no pairs of clean and noisy data are required. Rather, you only need separate samples of $p(x)$ and $p(n)$. This allows for a more flexible and general framework that does not require retraining of diffusion models for different tasks and works well on out-of-distribution data. In many practical scenarios — like ultrasound, radar, or audio processing — structured noise interference can be recorded in isolation (e.g., by recording the sensor without pointing to the signal of interest). To showcase this, we **added an experiment on an in-vivo medical ultrasound dataset** of the heart, which often experiences structured noise artifacts known as haze. In this instance, the haze artifact was separately acquired using an ultrasound probe. Subsequently, two diffusion priors were learned and leveraged in the proposed joint sampling method to remove the haze (noise) artifact. We have added a paragraph in the introduction to clarify this point. This experiment demonstrates that the method extends beyond purely synthetic setups.
>
> > _In eq. (8) you give separate expressions for $\nabla\_{x\_{t}}$ and $\nabla\_{n\_{t}}$, but in eq. (7) you should have $\nabla\_{x\_{t}, n\_{t}}$?_
>
> We agree with the reviewer that the relationship between eq. (7) and eq. (8) was not clearly defined. We have therefore rewritten eq. 8 as follows:
>
> $$
> \\nabla\_{\\mathbf{x}\_t, \\mathbf{n}\_t} \\log p(\\mathbf{x}\_t, \\mathbf{n}\_t \\mid \\mathbf{y}) =
> \\begin{bmatrix}
>     \\nabla\_{\\mathbf{x}\_t} \\log p(\\mathbf{x}\_t, \\mathbf{n}\_t \\mid \\mathbf{y}) \\\\[4pt]
>     \\nabla\_{\\mathbf{n}\_t} \\log p(\\mathbf{x}\_t, \\mathbf{n}\_t \\mid \\mathbf{y})
> \\end{bmatrix}
> \\approx
> \\begin{bmatrix}
>     s\_\\theta^*(\\mathbf{x}\_t, t) + \\lambda \\nabla\_{\\mathbf{x}\_t} \\log p(\\mathbf{y} \\mid \\mathbf{x}\_t, \\mathbf{n}\_t) \\\\[4pt]
>     s\_\\phi^*(\\mathbf{n}\_t, t) + \\kappa \\nabla\_{\\mathbf{n}\_t} \\log p(\\mathbf{y} \\mid \\mathbf{x}\_t, \\mathbf{n}\_t)
> \\end{bmatrix}
> $$
>
> which breaks up the vector differential into its signal and noise components, which now can be substituted in eq. (7).
>
> > _Why PSNR and SSIM as the only evaluation metrics, would it not be relevant to also include perceptual metrics such as LPIPS and FID?_
>
> We have included **LPIPS** to also include perceptual scores and explore both ends of the perception-distortion tradeoff.
>
> > _Is there really no diffusion-based method that could be relevant to include as a baseline? For example, what is the key difference between the proposed method and "Image Restoration with Mean-Reverting Stochastic Differential Equations" (Luo et al., ICML 2023)?_
>
> We have now **added a comparison with popular diffusion posterior sampling methods**, without the specific use of a noise prior, as a way to compare against diffusion models. The referenced paper by the reviewer is interesting, however, there are a few relevant differences with our method. Their method requires pairs of clean and noisy data for training their score networks. This is a big constraint on the dataset curation, as we cannot independently acquire datasets for signal and noise anymore as they need to be paired. Additionally, this supervised aspect of their method loses the advantages of unconditional score priors, such as robustness to out-of-distribution data (not trained to a specific inverse problem) and the flexibility to use them in other inverse problems without retraining. Lastly, our method fits in the Bayesian posterior sampling framework, which can leverage further improvements of posterior sampling techniques with diffusion models. **We have now included this reference in our introduction.**

---

> > ### Author Response · Authors · 2025-01-14
> > **Part 2**
> >
> > > _Why are there no quantitative results reported for the experiments in Figure 4 or Figure 5?_
> >
> > We have now included quantitative results for the experiments in Figure 4 and Figure 5.
> >
> > > _The formatting of Figure 4 and 5 is perhaps not ideal, maybe combine into sub-figures of a single figure?_
> >
> > We have now addressed the inconsistent offset in vertical spacing between Fig. 4 and 5.
> >
> > > _Figure 1 is never mentioned in the text? Eq. (1) is missing a "="? Should it not be "+" instead of "-" in eq. (3)?_
> >
> > We thank the reviewer for pointing out these typos, which now all have been rectified.
> >
> > > _I think it could be helpful to add a short paragraph to the start of Section 3, clarifying what will be covered in the following subsections._
> >
> > We have now included a short summary of the discussed topics in Section 3.
> >
> > > _Inconsistent use of e.g. "in equation 7" and "in (9)" (Table 1 caption) when referencing equations. Also, "Substitution of the VI estimate (12) into equation (11)" in Section 3.2.1._
> >
> > We have now changed all to use of `\eqref` whenever referencing equations.
> >
> > > _"Song et al. provide recommendations for choosing the variance of the VI Song et al. (2023)" in Section 3.2.1, odd citation formatting. Section 4, "Whang et al. propose to use normalizing flows to model both the data and the noise distributions Whang et al. (2021)", also odd citation formatting._
> >
> > We now have fixed all non-standard citation formatting, with proper use of `\citet` and `\citep`.
> >
> > > _Multiple "(Song et al., 2020)" citations should actually be "(Song et al., 2021b)", for the projection method, or?_
> >
> > See Song et al, 2020 eq 14 and Appendix I.4 where they first introduced a way of utilizing diffusion models for inverse problems solving. However, we do understand the confusion as wel also cite Song et al., 2021b later on the paragraph which goes into more detail (same author different paper). **We have now clarified this appropriately in the text (Appendix A.2).**
> >
> > > _Figure 2 caption, "to the baselines2: †FLOW, ‡GAN, and §BM3D", but there are no footnotes anywhere?_
> >
> > The footnote was accidentally moved to the Appendix, which we fixed and also include in the main body now.

---

> > > ### Comment · Reviewer_DXv6 · 2025-01-20
> > >
> > > Thank you for the response!
> > >
> > > I have read the other reviews and all author responses.
> > >
> > > The authors have provided a really solid and thorough rebuttal, and I definitely think that the paper has been improved after the updates (especially the added ultrasound experiment, and the restructuring of Section 6).
> > >
> > > I only have two questions on the rebuttal:
> > > - _"Added a paragraph in the introduction to clarify the need for samples of the noise distribution and how this can be obtained in practice"_, I don't seem to be able to find this?
> > > - Results for DPS are missing in Table 3 and Table 4 in the appendix?
> > >
> > >
> > > Provided that no major remaining issue is raised by the other reviewers, I plan to recommend accept.

---

> > > > ### Author Response · Authors · 2025-01-20
> > > >
> > > > We appreciate your kind words about the improvements to the paper and agree it has improved substantially with the feedback of the reviewers.
> > > >
> > > > > _Missing paragraph in the introduction_
> > > >
> > > > Thank you for catching this! We have updated the introduction in the revised submission to explicitly clarify the practical approach for obtaining samples of the noise distribution. The newly added paragraph reads:
> > > > _"While it is often challenging to derive analytical models for these structured noise distributions, samples can be practically obtained through simulation or by isolating noise in the absence of the signal of interest."_
> > > >
> > > > > _Results for DPS are missing in Table 3 and Table 4 in the appendix?_
> > > >
> > > > Indeed! These two tables are also updated with DPS scores in the revised version. Notably, DPS achieves its strongest performance in this experiment, likely due to the noise in this experiment being closest to Gaussian and exhibiting limited structure, aligning well with the assumptions of DPS. Visually it is still being outperformed by the proposed method, as is also reflected in the LPIPS metric.
> > > >
> > > > We hope this addresses your concerns fully. Thank you again for your review and insightful comments.

---

> > > > > ### Comment · Reviewer_DXv6 · 2025-01-30
> > > > >
> > > > > Thank you, I will recommend accept.

---

### Review · Reviewer_u795 · 2024-12-22

**Summary Of Contributions:**

This paper proposes to extend the diffusion priors of signals to structured noises in inverse problems such as denoising and compressed sensing. This idea is formulated as jointly inferring the signals x and noises n from the (corrupted) observation y from the posterior p(x, n|y) \propto p(y|x, n)p(x)p(n). After constructing a reverse-time SED for the joint conditional diffusion process {x_t, n_t|y}, the focus is shifted to the score function \nabla_{x_t, n_t} \log p(x_t, n_t|y), which can be decomposed based on the factorization of p(x_t, n_t|y). This formulation extends the previous ones with tractable noise distributions. The authors then apply this extension to three existing methods for inverse problems: Pseudo-Guided Diffusion Models, Diffusion Posterior Sampling, and projection, each has their own way of approximating p(y|x_t, n_t). Due to the parallelism between x and n, all three extensions are quite straight-forward. Experiments include (1) removing MNIST digits (as corruption) from CelebA signals, (2) sinusoidal noise with compressed sensing, (3) removing sinusoidal noise, (4) generalizing to out-of-distribution data/noise, (5) deraining FFHQ. The proposed method outperforms GAN, Flow-based method, block-matching and 3D filtering algorithm, and LASSO.

**Audience:**

Yes

**Claims And Evidence:**

Yes

**Requested Changes:**

I wonder if the authors would like to provide a comparison with the original Pseudo-Guided Diffusion Models, Diffusion Posterior Sampling, and projection in the same dataset i.e. CelebA, with both simple and richly structured noises.

**Strengths And Weaknesses:**

Strengths:
- The proposed extension is simple and straight-forward.
- The experiments on out-of-distribution data/noise are interesting.
- The comparison with non-Diffusion methods look convincing.

Weakness:
- The proposed method needs samples from noise distributions, which may be a constraint.

---

> ### Author Response · Authors · 2025-01-14
>
> First of all we would like to thank the reviewer for their insightful comments and time dedicated to review our work.
>
> In the following we address the comments of the reviewer.
>
> > _The proposed method needs samples from noise distributions, which may be a constraint._
>
> This is a valid point. Our method requires separate samples of each distribution ($x$ and $n$). The benefit, however, of this method is that no pairs of clean and noisy data are required. Rather, you only need separate samples of $p(x)$ and $p(n)$. This allows for a more flexible and general framework that does not require retraining of diffusion models for different tasks and works well on out-of-distribution data. In many practical scenarios — like ultrasound, radar, or audio processing — structured noise interference can be recorded in isolation (e.g., by recording the sensor without pointing to the signal of interest). To showcase this, we **added an experiment on an in-vivo medical ultrasound dataset** of the heart, which often experiences structured noise artefacts known as haze. In this instance, the haze artifact was separately acquired using an ultrasound probe. Subsequently, two diffusion priors were learned and leveraged in the proposed joint sampling method to remove the haze (noise) artefact. We have added a paragraph in the introduction to clarify this point. This experiment demonstrates that the method extends beyond purely synthetic setups.
>
> > _I wonder if the authors would like to provide a comparison with the original Pseudo-Guided Diffusion Models, Diffusion Posterior Sampling, and projection in the same dataset i.e. CelebA, with both simple and richly structured noises._
>
> Thank you for this suggestion. We indeed realize it might not be so trivial to see that current diffusion samplers with a Gaussian noise assumption fail on the structured noise experimenets. We now **compare in all our experiments (including the CelebA experiment) with three common diffusion sampling methods**, both with the existing Gaussian noise assumption and with the proposed joint sampling framework (which assumes structured noise). We do indeed see that the proposed method outperforms the Gaussian noise assumption in all experiments.

---

### Review · Reviewer_Z8d2 · 2025-01-02

**Summary Of Contributions:**

This paper proposes a generalised method for solving the inverse problem
$$ y = Ax + n ,$$
where we now assume that $n$ is possibly not Gaussian noise.
This is novel to current literature which currently assumes either Gaussian or Poisson noise models.

Their model of the noise is to assume that the diffusion noise on $x$ and the observation noise $n$ are independent.
By doing so we have the relation
$$ p_{X,N} ( x,n|y) \propto  p_{Y|X, N} (y|x,n) p_{X}(x) p_{N}(n) $$
where we can model both the signal variable $x$ and noise variable $n$ with a diffusion model.
The authors derive the joint SDE for the reverse sampling process, where the score along the diffusion path must be estimated.
The authors show how three current methods for score estimation along the diffusion path may be used with their method.
Experiments on Celeb-A with MNIST noise are given, along with other synthetic data examples.

**Audience:**

Yes

**Broader Impact Concerns:**

none.

**Claims And Evidence:**

Yes

**Requested Changes:**

I request that the authors add baselines of comparing their method to just assuming that $n$ is Gaussian. It would be interesting if the authors could identify a non-synthetic example of where the standard assumption of Gaussianity on $n$ is a poor approximation, and the added flexibility from their methodology could improve performance on this.

I also request that the authors add additional experimental details in their appendix. Currently I do not understand how either score network was trained, the training schedule used, amount of sampling steps taken, the sampling schedule used in their diffusion process or the early stopping parameters used. For instance, for diffusion inverse problems with Gaussian noise, the guidance strength is currently a sensitive hyper-parameter. In this work, there is a guidance strength in both the $x$ and $n$ directions. I cannot currently find these parameters in the text, but are required to run Algorithm 1.

**Strengths And Weaknesses:**

The main strength of this paper is the novelty of modelling the additive noise with a diffusion model. This is an interesting extension for diffusions for solving inverse problems as it should in theory generalise the applicability of diffusion models for inverse problems.
Generally, the paper is well presented.

Currently I do not understand how the score for the noise $n$ is trained. I believe there is an assumption that we can sample from the noise generating process, that is we have access to samples $(x,n)$. This is a current limitation for applying the method to realistic scenarios, as in practice we do not have these samples. I believe it would be highly beneficial if the authors could give an example in their computations section where the true noising process is not known and they can demonstrate an increase in performance with their method over assuming additive Gaussian noise.

Another limitation is the authors assume that $x$ and $n$ are independent. One realistic scenario for non-Gaussian noise $n$ could be that the data also contains heteroskedasticity. If the authors could comment if such an extension would be possible, this would make the current work a step closer to more applied use.

---

> ### Author Response · Authors · 2025-01-14
>
> First of all we would like to thank the reviewer for their insightful comments and time dedicated to review our work.
>
> In the following we address the comments of the reviewer.
>
> > _Currently I do not understand how the score for the noise n is trained. I believe there is an assumption that we can sample from the noise generating process, that is we have access to samples (x,n). This is a current limitation for applying the method to realistic scenarios, as in practice we do not have these samples._
>
> We appreciate the concerns about real-world applicability and clarifying how the noise score is trained. Our method requires separate samples of each distribution (x and n). The benefit of this method is that no pairs of clean and noisy data are required. Rather, you only need separate samples of $p(x)$ and $p(n)$. This allows for a more flexible and general framework that does not require retraining of diffusion models for different tasks and works well on out-of-distribution data. In many practical scenarios — like ultrasound, radar, or audio processing — structured noise interference can be recorded in isolation (e.g., by recording the sensor without pointing to the signal of interest). To showcase this, we **added an experiment on an in-vivo medical ultrasound dataset** of the heart, which often experiences structured noise artefacts known as haze. In this instance, the haze artifact was separately acquired using an ultrasound probe. Subsequently, two diffusion priors were learned and leveraged in the proposed joint sampling method to remove the haze (noise) artefact. We have added a paragraph in the introduction to clarify this point. This experiment demonstrates that the method extends beyond purely synthetic setups.
>
> > _Another limitation is the authors assume that x and $n$ are independent. One realistic scenario for non-Gaussian noise n could be that the data also contains heteroskedasticity. If the authors could comment if such an extension would be possible, this would make the current work a step closer to more applied use._
>
> This is an interesting extension to our work. As currently we address inverse problems with structured noise, where we assume that the noise is independent of the signal. However, we agree that in many practical cases, the noise is not independent of the signal, e.g. as is the case in the dehazing ultrasound experiment. Our model empirically still achieved good results, suggesting that limited violations of this independence do not hamper the working of our model. We have added a paragraph in the discussion to discuss this limitation and potential future work and leave extensions of our measurement model to non-linear measurement functions and non-independend $p(x)$ and $p(n)$ to future work, which could potentially follow similar methodologies as proposed in the DPS paper.
>
> > _I request that the authors add baselines of comparing their method to just assuming that n is Gaussian. It would be interesting if the authors could identify a non-synthetic example of where the standard assumption of Gaussianity on n is a poor approximation, and the added flexibility from their methodology could improve performance on this._
>
> We have now **added comparisons with common diffusion posterior sampling methods** (projection, DPS, $\Pi$GDM) as baselines, without the specific use of an explicitly learned noise prior, i.e. with Gaussian noise prior as assumption. We have also added an experiment on medical ultrasound data, which experiences from a structured noise artifact known as haze. We show that the proposed method with learned noise prior outperforms the Gaussian noise assumption in these experiments.
>
> > _I also request that the authors add additional experimental details in their appendix. Currently I do not understand how either score network was trained, the training schedule used, amount of sampling steps taken, the sampling schedule used in their diffusion process or the early stopping parameters used._
>
> We have now added experimental details and specific hyperparameter used in a summary table in the Appendix. Furthermore, we link to our code repository which lists the exact config files and models on HuggingFace Hub to reproduce the results (link will follow on GitHub repo when linked after review).

---

### Author Response · Authors · 2025-01-14
**Summary of author revisions**

We would like to thank all the reviewers for their time reading the paper and providing valuable feedback.

We have summarized some of the more substantial proposed changes below. We are encouraged to see the reviewers believe this work: _"is an interesting extension to diffusion"_ and _"well-presented"_ (reviewer **Z8d2**), _"is simple and straight-forward"_ (reviewer **u795**), _"is interesting and relevant"_ (reviewer **DXv6**). All changes in the revised paper have been **highlighted in green**. We have addressed all the questions posed by the reviewers and made the following main changes to the manuscript:

- Main request of the reviewers was to showcase **a real-world example of structured noise**. Therefore, we have included an experiment on medical ultrasound data, which experiences from a structured noise artefact known as haze.
- Second, we have added **comparisons with popular diffusion posterior sampling methods** (projection, DPS, $\Pi$GDM), without the specific use of a noise prior, as a way to compare against diffusion models.
- Restructured Sections 3 (methods) and 6 (experiments) for better readability. Some experiments are moved to the Appendix.
- Added a more complete set of quantitative results to the Appendix, as well as included **LPIPS as a perceptual metric**.
- Fixed typos and formatting issues throughout the manuscript.
- Added a paragraph in the introduction to clarify the need for samples of the noise distribution and how this can be obtained in practice.

---

### Decision · Action_Editor_E1Zn · 2025-03-13

**Recommendation:** Accept as is

**Comment:**

See above.

**Audience:**

Yes.

**Claims And Evidence:**

In this paper the authors propose to solve complex inverse problem using diffusion models.
Unlike approach like Diffusion Posterior Sampling, here the noise is not assumed to be Gaussian but instead to be structured.
Therefore it is also modelled by the diffusion processed.
The methodology is new, the experiments are convincing and the paper is well-written.
All the reviewers recommend acceptance and the authors have subsequently improved the paper in the rebuttal period.